# Comparison of Selected Immune and Hematological Parameters and Their Impact on Survival in Patients with HPV-Related and HPV-Unrelated Oropharyngeal Cancer

**DOI:** 10.3390/cancers13133256

**Published:** 2021-06-29

**Authors:** Adam Brewczyński, Beata Jabłońska, Agnieszka Maria Mazurek, Jolanta Mrochem-Kwarciak, Sławomir Mrowiec, Mirosław Śnietura, Marek Kentnowski, Zofia Kołosza, Krzysztof Składowski, Tomasz Rutkowski

**Affiliations:** 1I Radiation and Clinical Oncology Department of Maria Skłodowska-Curie National Research Institute of Oncology, 44-102 Gliwice Branch, Poland; Adam.Brewczynski@io.gliwice.pl (A.B.); marek.kentnowski@io.gliwice.pl (M.K.); Krzysztof.Skladowski@io.gliwice.pl (K.S.); Tomasz.Rutkowski@io.gliwice.pl (T.R.); 2Department of Digestive Tract Surgery, Medical University of Silesia, 40-752 Katowice, Poland; mrowasm@poczta.onet.pl; 3Centre for Translational Research and Molecular Biology of Cancer of Maria Skłodowska-Curie National Research Institute of Oncology, 44-102 Gliwice Branch, Poland; agnieszka.mazurek@io.gliwice.pl; 4The Analytics and Clinical Biochemistry Department of Maria Skłodowska-Curie National Research Institute of Oncology, 44-102 Gliwice Branch, Poland; Jolanta.Mrochem-Kwarciak@io.gliwice.pl; 5Tumor Pathology Department of Maria Skłodowska-Curie National Research Institute of Oncology, 44-102 Gliwice Branch, Poland; Miroslaw.Snietura@io.gliwice.pl; 6Department of Biostatistics and Bioinformatics of Maria Skłodowska-Curie National Research Institute of Oncology, 44-102 Gliwice Branch, Poland; zofia.kolosza@io.gliwice.pl

**Keywords:** oropharyngeal cancer, Human Papillomavirus (HPV), immune status, hematological parameters, radiotherapy

## Abstract

**Simple Summary:**

This is a research article on oropharyngeal cancer (OPC). The aim of the study was to assess and compare basic immune parameters and ratios in patients with Human Papilloma Virus (HPV)+ and HPV− OPC, before and after radiotherapy (RT) or chemoradiotherapy (CRT), and to investigate their impact on overall survival (OS) and disease-free survival (DFS). The higher neutrophil-lymphocyte ratio (NLR) and systemic immune inflammation (SII) are significant adverse prognostic factors for HPV+ OPC patients, because they are significantly associated with both inferior OS and DFS in this group, whereas the higher platelet cells (PLT) count is significant adverse prognostic factor for HPV− OPC patients, because it is significantly associated with inferior OS and DFS in this group. This study confirmed that determination of HPV etiology as well as analysis of various hematological and immune parameters should be a standard management in OPC patients in order to properly treat them for improved prognosis.

**Abstract:**

Several immune and hematological parameters are associated with survival in patients with oropharyngeal cancer (OPC). The aim of the study was to analyze selected immune and hematological parameters of patients with HPV-related (HPV+) and HPV-unrelated (HPV−) OPC, before and after radiotherapy/chemoradiotherapy (RT/CRT) and to assess the impact of these parameters on survival. One hundred twenty seven patients with HPV+ and HPV− OPC, treated with RT alone or concurrent chemoradiotherapy (CRT), were included. Patients were divided according to HPV status. Confirmation of HPV etiology was obtained from FFPE (Formalin-Fixed, Paraffin-Embedded) tissue samples and/or extracellular circulating HPV DNA was determined. The pre-treatment and post-treatment laboratory blood parameters were compared in both groups. The neutrophil/lymphocyte ratio (NLR), platelet/lymphocyte ratio (PLR), monocyte/lymphocyte ratio (MLR), and systemic immune inflammation (SII) index were calculated. The impact of these parameters on overall (OS) and disease-free (DFS) survival was analyzed. In HPV+ patients, a high pre-treatment white blood cells (WBC) count (>8.33 /mm^3^), NLR (>2.13), SII (>448.60) significantly correlated with reduced OS, whereas high NLR (>2.29), SII (>462.58) significantly correlated with reduced DFS. A higher pre-treatment NLR and SII were significant poor prognostic factors for both OS and DFS in the HPV+ group. These associations were not apparent in HPV− patients. There are different pre-treatment and post-treatment immune and hematological prognostic factors for OS and DFS in HPV+ and HPV− patients. The immune ratios could be considered valuable biomarkers for risk stratification and differentiation for HPV− and HPV+ OPC patients.

## 1. Introduction

It has been proven that significant alterations in the immunological system (IS) are observed in cancer patients. The body’s specific immune response to a cancer leads to various changes in levels of basic immune parameters such as number of white blood cells (WBCs), circulating lymphocytes (CLCs), circulating neutrophils (CNCs), circulating monocytes (CMCs) and platelet cells (PLTs) [1,2,3]. These parameters are easy to assess in the routine peripheral blood morphology. Additionally, based on the basic abovementioned parameters, the following immune ratios can be calculated: neutrophil/lymphocyte (NLR), platelet/lymphocyte (PLR), and monocyte/lymphocyte (MLR). It is known that they strongly influence survival in cancer patients. This also applies to patients with squamous head and neck cancer, including oropharyngeal cancer (OPC) [1,2,3].

OPC can be associated with a typical risk factor such as smoking and alcohol abuse or with Human Papillomavirus (HPV) infection. HPV-related (HPV+) OPC is fairly responsive to radiotherapy (RT) or chemoradiotherapy (CRT) and has a better prognosis than HPV− [4,5].

Because of the different nature of HPV+ OPC, the current 8th edition of the American Joint Committee on Cancer *AJCC Staging Manual* reflected HPV infection status in determining the clinical stage of OPC [4,5]. It is interesting to note whether the difference in OPC etiology, considering HPV status, causes differences in the immune status and whether it affects the prognosis and patient survival. There are various reports regarding immune alterations in HPV+ OPC cancer in the literature [6].

The aim of the study was to assess and compare basic immune parameters and ratios in patients with HPV+ and HPV− OPC, before and after RT or CRT, and to investigate their impact on overall survival (OS) and disease-free survival (DFS).

## 2. Materials and Methods

### 2.1. Patients

The analysis included 127 adults with OPC treated at I Radiation and Clinical Oncology Department of Maria Sklodowska-Curie National Research Institute of Oncology, Gliwice Branch, Poland. There were 87 (68.5%) men and 40 (31.5%) women of the mean age of 60.62 ± 8.54 years (range: 30–80 years) in the studied group. The inclusion criteria were as follows: primary OPC (T1–T4, N0–N3, M0), age > 18 years, radical RT or CRT as sole and definitive treatment. Exclusion criteria included: cancer recurrence, initial surgery, incomplete demographic and/or clinical data.

The detailed analysis of clinicopathological parameters (age, gender, tumor grading and staging, smoking) in OPC patients is presented in Table 1.

### 2.2. Study Design

#### 2.2.1. The Information on Grant and Ethical Standards

This study was supported by a grant from the National Centre of Research and Development, Poland (grant TANGO2/340829/NCBR/2017). All procedures performed in studies involving human participants were in accordance with the ethical standards of the institutional research committee (the Bioethics Committee at Maria Skłodowska-Curie National Research Institute of Oncology, Gliwice Branch, KB/43018/13) and with the 1964 Helsinki Declaration and its later amendments or comparable ethical standards. Informed consent was obtained from all individual participants included in the study.

#### 2.2.2. Laboratory Blood Investigations and Analysis

The blood was obtained under standard conditions, the patients in fasting state, between 7:00 and 9:00, by means of a vacuum Becton Dickinson (Franklin Lakes, NJ, USA) system, in sample tubes with anticoagulant ethylenediaminetetraacetic acid (EDTA). The full blood count was determined using the Sysmex XN-2000 analyzer (Sysmex, Kobe, Japan).

The pre-treatment (0) and post-treatment (1) basic laboratory blood parameters, including hemoglobin (Hb) and ret-hemoglobin (RetHb) levels, red blood cell count (RBC), reticulocyte count (Ret), white blood cell count (WBC), circulating lymphocyte count (CLC), circulating neutrophil count (CNC), and circulating monocyte count (CMC) were compared in both groups and the impact of immune and hematological parameters on survival was analyzed. The neutrophil/lymphocyte ratio (NLR), platelet/lymphocyte ratio (PLR), and monocyte/lymphocyte ratio (MLR), and systemic immune inflammation index (SII) were calculated and correlated with survival in both groups. The SII index was calculated according to the following formula: SII = platelet counts × neutrophil counts/lymphocyte counts [7]. Patients were divided into two groups depending on the HPV status: HPV-negatives (HPV−) and HPV-positives (HPV+).

Additionally, the patients were divided into two subgroups according to the cut-off value of the mean NLR, MLR, PLR, and SII. The mean values of the NLR, MLR, and PLR among the entire study population, as well as HPV− and HPV+ patients, were set as the border value to divide high and low NLR, MLR, and PLR subgroups in order to perform statistical comparisons of clinicopathological findings between these subgroups. Clinicopathological factors were compared between these low and high immune ratio subgroups.

#### 2.2.3. Confirmation of the HPV Etiology

Confirmation of the HPV etiology was obtained from tissue material and/or extracellular circulating HPV DNA.

##### Tissue Material

Formalin-fixed paraffin-embedded tumor samples were examined for high-risk HPV (HR-HPV) infection using a double-check algorithm including immunohistochemical assessment of P16(INK4A) protein expression followed by detection of HR-HPV DNA in tumor tissue using real-time PCR. Only cases with both p16(INK4A) expression and HR-HPV DNA amplification were classified as truly HR-HPV-positive [7].

##### Analysis of cfHPV16 DNA in Plasma

Peripheral blood (12 mL) was collected into K3EDTA tubes (Becton Dickinson, Franklin Lakes, NJ, USA). Plasma was separated within an hour by double centrifugation at 300× *g* and 1000× *g*, both at 4 °C for 10 min. DNA was extracted (according to the manufacturer’s instructions) from 1 mL of plasma by the Genomic Mini AX Body Fluids kit (A&A Biotechnology, Gdynia, Poland). Each measurement consisted of a standard curve of three dilutions of plasmid construct containing HPV16 genome, negative control and a samples. For HPV16 detection, reaction was performed using primers and probe set for the HPV16 genome. PCR reactions were performed using the Bio-Rad CFX96 qPCR instrument (Bio-Rad Laboratories, Hemel Hempstead, UK). If HPV16 was found, its presence would be confirmed with a second independent DNA isolation.

#### 2.2.4. Histopathological Staging and Grading Classification

The stage of OPC was classified according to the 8th edition of the American Joint Committee on Cancer (AJCC) TNM classification system [4,5].

#### 2.2.5. Follow-Up

The median follow-up was 74.58 (0.1–165.58) months. Overall survival (OS) and disease-free survival (DFS) were analyzed in both groups.

The flowchart diagram of our study is presented in Figure 1.

### 2.3. Statistical Analysis

The categorical variables were presented as numbers and percentages. Continuous variables with normal distribution were expressed as the means and standard deviations. The Shapiro–Wilk test was used to determine statistical distribution in the analyzed patients. The Mann–Whitney *U* test was used to compare HPV+ and HPV− groups. The Wilcoxon test was used to compare pre- and post-treatment parameters in all patients and both HPV groups separately. Prevalence and frequency were expressed as number and percentage. Cox proportional-hazards models were used to estimate hazard ratios (HRs) for OS and DFS. Receiver operating characteristic (ROC) curve analysis was performed to determine the optimal cut-off values for prognostic factors related to DFS and OS. Youden’s index was selected as the approximate cut-off value for each parameter. The Kaplan–Meier curves were constructed for comparison of OS and DFS between the two groups.

The log-rank test was used to assess the equality of survival distributions across different strata. The hazard ratio for death among patients with HPV− and HPV+ was determined. A *p*-value of equal or less than 0.05 was considered to be statistically significant. The statistical analyses were performed using the Statistica^®^ software program, version 13.0 (StatSoft).

## 3. Results

### 3.1. General Characteristics

The general clinical characteristics of 127 patients is presented in Table 1.

Both groups were comparable regarding the age and gender structure. The mean age was 60.85 ± 7.48 (37–79) and 60.36 ± 9.67 (30–80) years in the HPV− and HPV+ groups, respectively (*p* = 0.745). The male gender was predominant in the both groups. There were 51 (75.1%) and 36 (61.0%) males in the HPV− and HPV+ groups, respectively (*p* = 0.133).

The tonsil was the most common OPC location in both groups, but the incidence of the tumor location was different depending the HPV status. The tonsil location was the most frequent (47 (79.7%)) in HPV+ patients, and this location was noted in only 44 (64.7%) HPV− patients (*p* = 0.010). The palate location was not observed in HPV+ patients, and this location was noted in 10 (14.7%) HPV− patients.

Concerning the histopathological findings, in both groups, G2 grading and T2/T3 staging were the most common. G2 grading was reported in 34 and 21 patients who were HPV− and HPV+, respectively. G3 grading was more frequently noted in HPV+ patients compared to HPV− ones (20.3% vs. 10.3%; *p* = 0.054). The tumor depth was similar in both groups. It should be emphasized that the HPV+ patients had the more advanced nodal status compared to HPV− patients (71.2% vs. 47.1% N2–3; *p* = 0.005).

The significantly higher regional advancement of the HPV+ tumors was associated with the difference in the treatment regimen in the both groups. RCT was significantly more frequently used in the HPV+ patients compared to HPV− patients (88.1% vs. 64.7%; *p* = 0.003).

### 3.2. Laboratory Results before and after Treatment

The basic laboratory results in all patients and in both HPV− and HPV+ groups before and after treatment are presented and compared in Appendix A.

Most pre-treatment (0) and post-treatment (1) laboratory results were comparable in both groups. Moreover, a decrease of the most parameters following RT/CRT was noted in our study. Only ret-hemoglobin (RetHb) increased after the treatment in HPV+ patients. Therefore, a significantly higher RetHb1 level was noted in HPV+ patients (35.01 ± 1.43) compared to HPV− patients (33.50 ± 3.38). A reticulocyte count (Ret) before (51.97 ± 24.96 vs. 60.56 ± 22.73, *p* = 0.052) and after treatment (51.20 ± 25.29 vs. 48.62 ± 23.88; *p* = 0.574) was comparable in HPV− and HPV+ groups, respectively, but the Ret decrease was significantly greater in HPV+ patients compared to HPV− ones (0.89 ± 29.78 vs. 12.30 ± 30.51, *p* = 0.044). A lower WBC0 was reported in HPV+ patients compared to HPV− participants (6.45 ± 1.91 vs. 7.15 ± 2.03, *p* = 0.048), while the difference in WBC1 between both groups was not statistically significant (4.35 ± 2.25 vs. 5.12 ± 2.29, *p* = 0.059). The WBC decrease was observed in both groups. CLC1 (*p* = 0.004) and CMC1 (*p* = 0.012) were significantly lower in HPV+ patients, while these pre-treatment parameters were comparable in HPV− and HPV+ groups (*p* = 0.842 for CLC0, *p* = 0.057 for CMC0). A significantly lower PLT1 count was recorded in HPV+ patients compared to HPV− ones (208.15 ± 65.18 vs. 250.31 ± 118.11, *p* = 0.016), while PLT0 was similar in HPV+ and HPV− groups (236.76 ± 57.94 vs. 256.93 ± 77.16, *p* = 0.103). It was associated with a greater PLT decrease in HPV+ patients (*p* = 0.218).

The NLR 0/1, MLR 0/1, PLR 0/1, and SII 0/1 were comparable in both groups. The results are presented in Appendix A.

### 3.3. Comparison of Clinical and Pathological Characteristics Depending on the Values of Immune Ratios in HPV− and HPV+ Patients

The differences between low and high pre-treatment immune ratios groups (NLR 0/1, MLR 0/1, PLR 0/1 and SII 0/1) were analyzed.

In the low NLR HPV− subgroup, the significantly greater incidence of tonsil location compared to the incidence in the high NLR subgroup was reported (73.9% vs. 45.5%; *p* = 0.039). In HPV+ patients, the incidence of tonsil location was comparable in both NLR subgroups (81.1% vs. 77.3%; *p* = 0.872, respectively). There was no significant difference in terms of the other clinicopathological parameters between the two NLR groups regardless of HPV status (Appendix A).

In the low MLR HPV− subgroup, the highest G3 grading was significantly more frequent compared to the high MLR subgroup (25.0% vs. 0.0%; *p* = 0.026). In a comparison of the determined grading, G2 was the commonest grading in both subgroups. In HPV+ patients, histological grading was comparable in both MLR subgroups. The other clinicopathological parameters were similar in both MLR subgroups (Appendix A).

In the low PLR HPV+ subgroup, smoking was observed significantly more frequently compared to the high PLR subgroup (43.6% vs. 15.0%; *p* = 0.042). This difference was not noted in HPV− patients. The other clinicopathological parameters were comparable in both PLR subgroups (Appendix A).

In HPV− patients, a higher initial BMI was noted in patients with the low SII compared to patients with the high SII (26.6 ± 4.8 vs. 24.1 ± 4.1 kg/m^2^; *p* = 0.028). In HPV+ patients, G3 grading was more frequent in the low SII group compared to patients with the high SII (47.8% vs. 9.1%; *p* = 0.021). Moreover, smoking was seen significantly more frequently in the low SII HPV+ subgroup compared to patients with the high SII (43.6% vs. 15.0%; *p* = 0.042) (Appendix A). All comparisons are presented in Appendix A.

### 3.4. Overall Survival and Disease-Free Survival Depending on HPV Status

OS and DFS in both HPV groups are presented in Table 2, Table 3, Table 4 and Table 5, Appendix A, and Figure 2, as well as Appendix A. OS and DFS depending on HPV status were assessed using Cox regression univariate (UVA) and multivariate analysis (MVA). The prognostic factors determined in UVA were confirmed and presented using Kaplan–Meier curves. Generally, HPV status was a very strong prognostic factor for OS and DFS in our patients. OS and DFS were significantly better in HPV+ patients compared to HPV− ones (*p* = 0.0008 and *p* = 0.0009, respectively) (Figure 2). The treatment strategy (RT/CRT) was not a prognostic factor for both HPV− and HPV+ patients, in UVA and MVA (*p* > 0.05). Therefore, the treatment regimen did not impact survival in our patients (Table 3 and Table 4).

#### 3.4.1. Overall Survival Depending on Pre-Treatment Parameters

##### HPV−

In UVA, poor prognostic factors for OS in HPV− patients were as follows: A higher CMC (HR 2.41, 95% CI 1.01–5.74, *p* = 0.048), a higher PLT (HR 2.53, 95% CI 1.07–5.99, *p* = 0.035), and a higher MLR (HR 3.73, 95% CI 1.71–8.15, *p* = 0.001). A higher WBC (HR 2.33, 95% CI 0.99–5.47, *p* = 0.053) marginally predicted inferior OS in HPV− patients.

In MVA, a higher CMC (HR 4.24, 95% CI 1.52–11.84, *p* = 0.006), a higher MLR (HR 3.83, 95% CI 1.46–10.02, *p* = 0.006), and a higher PLR (HR 3.60, 95% CI 1.15–11.31, *p* = 0.028) were poor prognostic factors for OS.

##### HPV+

In UVA, poor prognostic factors for OS in HPV+ patients were as follows: A higher WBC (HR 4.17, 95% CI 1.25–13.93, *p* = 0.020), a higher NLR (HR 4.76, 95% CI 1.29–17.57, *p* = 0.019), and a higher SII (HR 5.67, 95% CI 1.24–25.94, *p* = 0.025). A higher Ret was associated with higher OS (HR 0.22, 95% CI 0.06–0.82, *p* = 0.025),

In MVA, a higher CMC (HR 17.18, 95% CI 1.89–167.47, *p* = 0.012), and a higher SII (HR 11.10, 95% CI 2.02–60.97, *p* = 0.006) were poor prognostic factors for OS.

All described prognostic factors are presented in Table 2.

#### 3.4.2. Disease-Free Survival Depending on Pre-Treatment Parameters

##### HPV−

In UVA, poor prognostic factors for DFS in HPV− patients were as follows: A higher PLT (HR 2.77, 95% CI 1.18–6.52, *p* = 0.020), and higher PLR (HR 2.88, 95% CI 1.30–6.38, *p* = 0.009), whereas a higher RBC (HR 0.39, 95% CI 0.15–0.97, *p* = 0.043) predicted better DFS.

In MVA, a higher PLR (HR 2.74, 95% CI 1.02–7.41, *p* = 0.046) was a poor prognostic factor for DFS, whereas a higher RBC (HR 0.21, 95% CI 0.07–0.66, *p* = 0.008), a higher MLR (HR 0.20, 95% CI 0.06–0.62, *p* = 0.006) were associated with higher DFS.

##### HPV+

In UVA, poor prognostic factors for DFS in HPV+ patients were as follows: A higher NLR (HR 6.02, 95% CI 1.21–29.87, *p* = 0.028), and a higher SII (HR 8.48, 95% CI 1.04–68.98, *p* = 0.046), whereas a higher CLC (HR 0.19, 95% CI 0.05–0.77, *p* = 0.020), a higher CMC (HR 0.16, 95% CI 0.03–0.8, *p* = 0.026) predicted higher DFS.

In MVA, a higher Hb (HR 10.44, 95% CI 1.31–83.38, *p* = 0.027) was a poor prognostic factor for DFS, whereas a higher CLC (HR 0.17, 95% CI 0.03–0.97, *p* = 0.046) was associated with higher DFS. A higher CMC (HR 0.20, 95% CI 0.04–1.07, *p* = 0.060) marginally predicted higher DFS in HPV+ patients.

All described prognostic factors are presented in Table 3.

#### 3.4.3. Overall Survival Depending on Post-Treatment Parameters

##### HPV−

In UVA, poor prognostic factors for OS in HPV− patients were as follows: A higher PLR (HR 2.55, 95% CI 1.13–5.77, *p* = 0.024), and a higher SII (HR 3.54, 95% CI 1.28–9.80, *p* = 0.015). A higher Hb (HR 0.27, 95% CI 0.08–0.93, *p* = 0.037) predicted better OS in HPV−patients.

In MVA, higher PLT (HR 3.37, 95% CI 1.18–9.62, *p* = 0.023) and a higher SII (HR 4.69, 95% CI 1.23–17.97, *p* = 0.024) were poor prognostic factors for OS. A higher RBC (HR 0.32, 95% CI 0.12–0.86, *p* = 0.024), a higher WBC (HR 0.16, 95% CI 0.03–0.93, *p* = 0.042) were associated with better OS.

##### HPV+

In UVA, only a higher CLC (HR 8.70, 95% CI 2.26–33.54, *p* = 0.002) was a poor prognostic factor for OS in HPV+ patients.

In MVA, also a higher CLC (HR 11.37, 95% CI 2.61–49.64, *p* = 0.001) was a poor prognostic factor for OS in HPV+ patients.

All described prognostic factors are presented in Table 4.

#### 3.4.4. Disease-Free Survival Depending on Post-Treatment Parameters

##### HPV−

In UVA, poor prognostic factors for DFS in HPV− patients were as follows: A higher NLR (HR 3.16, 95% CI 1.18–8.50, *p* = 0.022) and a higher SII (HR 3.25, 95% CI 1.27–8.28, *p* = 0.014), whereas a higher Hb (HR 0.37, 95% CI 0.15–0.89, *p* = 0.027) and a higher RBC (HR 0.42, 95% CI 0.19–0.93, *p* = 0.033) were associated with better DFS.

In MVA, a higher PLR (HR 3.96, 1.46–10.77, *p* = 0.007) was a poor prognostic factor for DFS, whereas a higher RBC (HR 0.26, 95% CI 0.11–0.66, *p* = 0.004) was associated with higher DFS.

##### HPV+

In UVA, only a higher PLT (HR 4.26, 95% CI 1.06–17.07, *p* = 0.041) was a poor prognostic factor for OS in HPV−+ patients.

In MVA, also a higher PLT (HR 7.97, 95% CI 1.55–41.00, *p* = 0.013) was the only poor prognostic factor for DFS.

All described prognostic factors are presented in Table 5.

#### 3.4.5. Summary of the Analysis of OS and DFS in HPV− and HPV+ Patients

In summary, our study showed different pre-treatment and post-treatment parameters predicting OS and DFS in HPV− and HPV+ patients. HPV status was the strongest predictor for OS and DFS. Generally, there were more immune and hematological parameters predicting survival in HPV− patients compared to HPV+ participants. In HPV+ patients, a high pre-treatment WBC, NLR, and SII significantly correlated with reduced OS, whereas a high NLR and SII significantly correlated with reduced DFS. A higher pre-treatment NLR and SII were significant poor prognostic factors for both OS and DFS in the HPV+ group. These associations were not apparent in HPV− patients.

Thus, there are different pre-treatment and post-treatment immune and hematological prognostic factors for OS and DFS in HPV+ and HPV− patients.

A summary of differences regarding the impact of immune and hematological pre-treatment and post-treatment parameters on OS and DFS depending on HPV status is presented in Table 6.

## 4. Discussion

Both neutrophils and monocytes are derived from a myelocytic lineage, whereas lymphocytes are derived from a lymphoid lineage [1]. The association of high baseline myeloid-derived cells (neutrophils and monocytes) with poor clinical outcomes has been observed in various cancer types, including head and neck cancers and OPC. This concerns both the blood and tumor cell counts [1,8,9,10]. In Huang et al.’s study [1], both high pre-treatment CNC and CMC values independently predicted poor survival and disease control, whereas a high CLC was associated with better recurrence-free survival (RFS) and marginally better OS in HPV+ OPC patients. A similar association was not reported in HPV− OPC patients. It is proof that the host immune system significantly influences treatment outcome in HPV+ OPC individuals [1]. Also, it has been shown that pre-treatment anemia is an independent poor prognostic factor for survival in HPV+ OPC patients [11].

In recent years, various peripheral blood inflammation factors, such as counts of neutrophils, lymphocytes, monocytes, platelet cells, either as individual values or ratios, have been proposed as prognostic markers of head and neck squamous cell carcinoma (HNSCC) [12,13,14,15]. Therefore, we decided to analyze their impact on prognosis in our OPC patients considering the status of high-risk HPV.

In our study, PLT count was a significant poor prognostic factor for both OS and DFS survival in HPV− patients. This association of the higher PLT count with inferior survival in OPC patients is in accordance with observations of other authors. Shoultz-Henley et al. [13] evaluated associations between increased PLT and anemia and oncologic outcomes in OPC patients receiving concurrent CRT. The authors noted that locoregional control (LRC), freedom from distant metastasis (FDM), and OS were significantly decreased for patients with a pre-treatment PLT value of ≥350 × 10^9^ /L. Anemic patients demonstrated comparatively decreased LRC, FDM, and OS. Additionally, patients with simultaneous PLT elevation and anemia had significantly worse oncologic outcomes for LRC, FDM, and OS than those with anemia or platelet elevation alone or those with no alteration. It should be emphasized, that Shoultz-Henley’s study did not stratify patients according to HPV status [13]. Gorphe et al. [11] investigated the prognostic value of pre-treatment hematological parameters in patients with HPV+ OPC. In their study, Hb < 12 g/dL was associated with impaired OS and PFS, pre-treatment NLR > 5 was associated with decreased OS. Patients with NLR > 5 had a significantly higher rate of disease recurrence. The authors explained the association between a low Hb level and poor prognosis in OPC patients as follows: low Hb concentration might exacerbate the preexisting hypoxia that is often present in tumors by decreasing oxygen-carrying capacity and so hampering the response of tumor cells to cytotoxic therapy [11]. In our study, with HPV status stratification, we showed a significant association between the pre-treatment Hb level and DFS only in HPV− patients. A similar association was not shown for HPV+ patients. In another study, Ye et al. reported that pre-treatment NLR elevation and PLT > 248 × 10^9^ /L were promising predictors of prognosis in patients with operable HNSCC. Explanation of the association between increased PLT and poor survival in OPC patients is the theory that PLTs direct tumor cell growth, vascular invasion, hematogenous dissemination, immune system evasion, and creation of metastasis site [12]. There are scant publications regarding associations between PLT and prognosis in OPC patients, but second factor NLR is the most frequently studied immune parameter. In many publications, it has been noted that increased CNC and decreased CLC were correlated with poor prognosis in OPC patients. This is due to the proven fact that an increased CNC in an inflammatory microenvironment contributes to tumor angiogenesis. It induces the resistance to anti-vascular endothelial growth factor (anti-VEGF) therapy. Additionally, the decreased CLC plays important roles in inflammatory reaction against tumor. Therefore, increased NLR is an independent prognostic factor for survival in cancer patients [12]. Rachidi et al. [14] reported that the CNC and LCC are strong biomarkers for poor prognosis and the NLR is a strong predictor of OS in oral, pharyngeal, and laryngeal squamous cell cancers. Additionally, the authors noted that a higher CNC correlated with a lower CLC. In these authors’ study, a higher CNC was associated with shorter OS, whereas a higher CLC was associated with longer OS. Also, their study demonstrated a bigger magnitude of correlation between NLR > 4.39 and survival within the HPV+ group than that seen in the HPV− group [14]. This phenomenon was also observed in many studies, including our research.

Charles et al. [15] compared patients with oropharyngeal and non-oropharyngeal HNSCC. With univariate analysis, the authors demonstrated associations between NLR and RFS and OS in both sub-populations. Multivariable analysis showed that patients with an NLR  >  5 had shortened OS in both sub-populations but an NLR  >  5 only predicted RFS in oropharyngeal patients [15]. Kano et al. [16] in a study conducted on patients with oropharyngeal, hypopharyngeal, and laryngeal cancers, demonstrated that a high NLR > 1.92, high PLR, and low LMR were all significantly associated with decreased OS and DFS. Valero et al. [17] reported that an increased NLR > 1.35 was independently related to inferior DFS in patients with HNSCC. Selzer et al. reported that a high NLR > 5 is associated with inferior OS in locally advanced head and neck cancer patients who were treated with curative intent by primary RT alone, or by RCT [18]. Moon et al., in HNSCC patients who underwent definitive RCT, observed that a higher NLR was associated with shortened progression-free survival (PFS) and OS [19]. In our study, an NLR above 2.13 for OS and above 2.29 for DFS was a significant poor prognostic factor in HPV+ patients. Yao et al. in a study conducted on patients with nasopharyngeal cancer, showed that an NLR > 2.50 was significantly associated with inferior OS, distant metastasis-free survival (DMFS), and PFS [20]. Lu et al. reported that an NLR ≥ 2.28, LMR < 2.26, and PLR ≥ 174 were significantly associated with a shorter OS, and an NLR ≥ 2.28 was significantly associated with a shorter PFS in patients with nasopharyngeal cancer [21].

The SII was the second independent prognostic factor for both OS and DFS in HPV+ patients in our study. The significant prognostic SII cut-off in the HPV+ group was >448.60 for OS and 462.58 for DFS. The SII as a prognostic factor for survival in cancer patients is less frequently described. Gao et al. described the SII, NLR, and PLR as independent prognostic factors for OS and DFS in patients with surgically resected esophageal squamous cell carcinoma (ESCC). The optimal cut-off values for the prediction of survival were 479.72 for the SII, 2.27 for NLR, 117.07 for PLR, and 0.19 for MLR [22]. Thus, the values of significant factors (NLR, SII) were similar to our findings. Among another immune ratios, our study showed only the statistical significance of MLR > 0.425 for OS in the HPV− group, and PLR > 173.2 for DFS in HPV− patients. The statistical analysis did not show significant associations for the other ratios, but observations of tendency within these parameters were comparable with the literature data. The mechanism of the SII contribution to a worse prognosis in patients with solid cancer is still unclear. There are some theories explaining the prognostic significance of the SII. According to the first hypothesis, the CNC expands both in the tumor microenvironment and systemically, and it is known that it is associated with poor prognosis in cancer patients [22,23]. The CNC may activate endothelium and parenchymal cells to enhance circulating tumor cell adhesion for distant metastasis [22,24]. The second theory is that PLTs may act as protective “cloaks” for circulating tumor cells (CTCs), protecting them from immune destruction. Additionally, PLTs and endothelial cell adhesion proteins may facilitate metastasis by augmenting tumor cell extravasation [22,25]. Thirdly, tumor-infiltrating lymphocytes (TILs) are associated with better response to cytotoxic treatment and prognosis in cancer patients [22,26,27]. The CLC can also secrete several cytokines, such as IFN-γ and TNF-α, in order to block tumor growth and improve the prognosis of cancer patients [22,28]. According to Gao et al. [22] and in our opinion, the SII should be a more objective marker, because it reflects the balance between host inflammatory and immune response status better than all the other immunological ratios, such as the NLR, PLR, and MLR. In our opinion, the SII is the best systemic inflammatory marker, superior to immune ratios (NLR, PLR, and MLR) and singular hematological parameters (CNC, CLC, CMC, and PLT). The SII involves three singular parameters (CNC, PLT, and CLC), more than the others (NLR, PLR and MLR), which include only two singular parameters. The more parameters we consider simultaneously, the better the parameter will reflect the systemic immune response.

The SII in the role of an adverse prognostic factor was described in various cancers as follows: oral squamous cell carcinoma [28], esophageal squamous cell carcinoma [22,29,30,31], colorectal cancer [32,33], lung cancer [34], pancreatic cancer [35], prostate cancer [36], hepatocellular cancer [37], gastric cancer [38], bladder cancer [39], renal cancer [40], cervical cancer [41], and breast cancer [42], but there is no report regarding the SII in OPC. To our knowledge, the present research is the first study on the SII in OPC patients in the worldwide literature, additionally with stratification based on HPV status. Association of the SII with OS and DFS was comparable to the NLR in our patients. Both ratios were significant poor prognostic factor for OS and DFS in HPV+ OPC patients, without such an association in the HPV− group.

Huang et al. [1] compared prognostic significance in HPV− and HPV− OPC patients. The authors noted that HPV+ OPC patients with a higher CNC, a higher CMC, and a lower CLC had inferior survival and an increased risk of disease recurrence. They did not show a similar association in HPV− patients. This observation only partially correlates with our results, because our study, conducted on 127 patients, showed a significant association between a lower CMC and CLC and inferior DFS in HPV+ patients, but did not show any correlation in the HPV− group similar to Huang’s report conducted on 510 adults. This phenomenon requires further observation in a larger patient group.

Generally, our study showed significantly better OS and DFS in HPV+ patients compared to HPV− ones (*p* = 0.0008 and *p* = 0.0009, respectively). Thus, HPV status was a very strong prognostic factor for OS and DFS. Survival of HPV+ patients was better despite the higher regional disease advancement. These results are in full accordance with the literature data. Patients with HPV-related OPC have a better prognosis and longer survival compared to patients without HPV-related OPC with typical risk factors (smoking, alcohol abuse) [43,44]. A better prognosis was also observed in HPV+ patients with more advanced OPC with lymph node involvement [5,6]. Moreover, the HPV+ OPC is more responsive to radiotherapy (RT) and chemoradiotherapy (CRT) [4,5,45]. It allows for treatment de-escalation in HPV+ patients [45,46,47]. In our opinion, this strong impact of HPV status on survival is associated with the presence of different prognostic factors in HPV− and HPV patients. The better survival in HPV+ OPC patients is associated with a greater locoregional control, higher sensitivity to radiation, or better radio-sensitization with the use of cisplatin [48]. The association between the superior survival of HPV+ OPC patients and the administered therapy is unclear. According to numerous authors, tumor HPV status is a strong and consistent determinant of better survival, regardless of treatment strategy (surgery, radiation therapy, concurrent CRT, or induction chemotherapy plus concurrent CRT) with five year survival rates among HPV+ patients of approximately 75 to 80%, versus 45 to 50% among HPV− patients [48,49,50,51,52].

Our patients were treated under standard department protocols definitively with radiotherapy (stage I-II) and chemoradiotherapy (stage III-IV). The difference in treatment strategy was determined by tumor stage rather than HPV status. The fact that more patients with HPV-related tumors presented more advanced stages was associated with a different tumor biology and clinical outcome. There was a significantly higher number of more advanced N2–3 tumors in HPV+ patients compared to HPV− ones. RT was significantly more frequent in N0–1 tumors, whereas CRT was more common in N2–3 tumors in an analysis of all patients together. The greater nodal involvement is characteristic for HPV+ OPC patients, and it has been confirmed by numerous studies. Moreover, according to the literature, typically HPV OPC presents in a younger, healthier population with a different set of risk factors and good prognosis for survival. Moreover, the majority of analyses showed that patients with HPV+ tumors had significantly better responses to treatment than those with HPV− tumors. HPV− OPC patients are usually older, with numerous comorbidities [43,53,54]. Therefore, in HPV− OPC patients CRT was not possible due to a general status and comorbidities, and accelerated RT was used in this patient group. The aim of our study was a comparison of immune and hematological parameters and their impact on survival in all HPV− and HPV+ OPC patients. Exclusion of any treatment strategy would reduce the possibility of a real clinical evaluation and would distort conclusions relevant to clinical practice. After all, total effective treatment strategy in clinical practice includes both RT and CRT use depending on the tumor staging. In addition, we compared OS and DFS depending on immune and hematological parameters in HPV− and HPV+ groups separately (not between HPV− and HPV+ patients). Moreover, the treatment strategy in low and high NLR, MLR, PLR, and SII groups in both HPV− and HPV+ patients was comparable (*p* > 0.05). Thus, treatment strategy had no impact the final results. Additionally, the impact of RT and CRT on survival has been assessed in univariate (UVA) and multivariate (MVA) Cox analysis. In both UVA and MVA, for HV- and HPV+ patients, the treatment strategy was not a significant prognostic factor. There are studies with a comparative analysis of survival depending on various parameters in head and neck cancer patients despite the statistical difference of the treatment strategies between compared groups in the literature [15,31,55].

To our knowledge, there is only one study regarding a similar topic. In this study, Huang et al. [1] investigated the prognostic value of the pre-treatment circulating neutrophil count (CNC), circulating monocyte count (CMC), and circulating lymphocyte count (CLC) in HPV− and HPV+ OPC patients. Although there are numerous reports regarding the prognostic role of the NLR, MLR, PLR, SII in various cancers, including OPC, there are no reports with a detailed and comprehensive comparison of these parameters between HPV− and HPV+ patients as well as analysis of their impact on survival in both HPV− and HPV+ patients. Our study included a detailed and comprehensive comparative analysis of numerous immune and hematological parameters. Our study is a Polish/Central European voice in the discussion regarding the prognostic role of hematological and immune parameters in HPV− and HPV+ OPC patients. It can be used in a further meta-analysis on this subject. So far, there are only a few original reports in this field, and there is no meta-analysis summarizing all cohort studies. Our study presents simple and widely available blood parameters that may be used in the clinical practice. Taking into account all the above mentioned arguments, the novelty of our study is considerable.

The single center observation and retrospective analysis of a prospectively collected database are limitations of our study. A prospective randomized multi-center study is needed to understand the biology of our observation and potentially to identify new therapeutic targets based on our findings.

## 5. Conclusions

The higher NLR and SII are significant adverse prognostic factors for HPV+ OPC patients because they are significantly associated with both inferior OS and DFS in this group, whereas the higher PLT is a significant adverse prognostic factor for HPV− OPC patients because it is significantly associated with inferior OS and DFS in this group. Further studies are needed in order to validate our findings. The knowledge of differences in immune parameters between HPV− and HPV+ OPC patients can be useful for the identification of new targeted immunotherapy in the HPV+ OPC treatment.

## Figures and Tables

**Figure 1 cancers-13-03256-f001:**
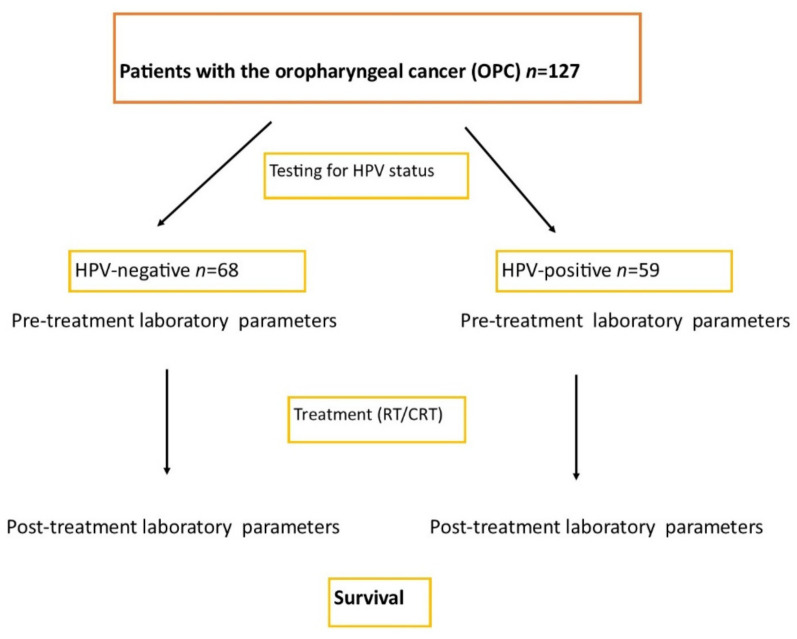
Flowchart diagram for the study.

**Figure 2 cancers-13-03256-f002:**
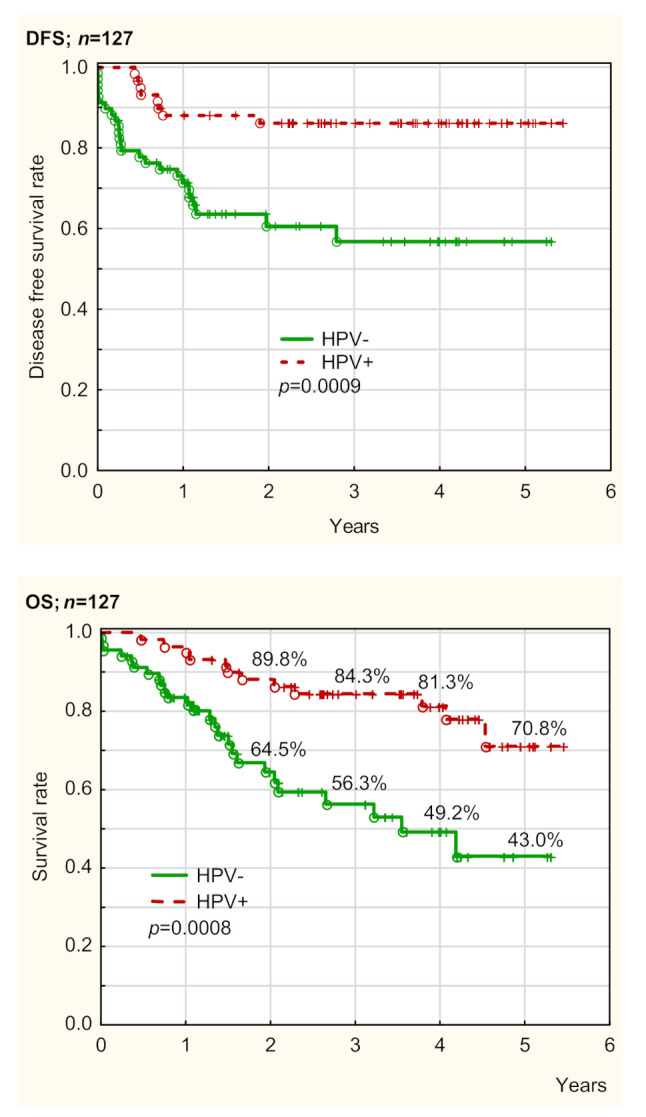
Comparison of overall survival (OS) and disease-free survival (DFS) between Human Papilloma Virus (HPV)− and HPV+ groups.

**Table 1 cancers-13-03256-t001:** The patients’ general clinicopathological characteristics.

Feature	All *n* = 127	HPV(−) *n* = 68	HPV(+) *n* = 59	*p*
Demographic characteristics				
Age	60.62 ± 8.54 (30–80)	60.85 ± 7.48 (37–79)	60.36 ± 9.67 (30–80)	0.745
Male	87(68.5%)	51 (75.1%)	36 (61.0%)	0.133
Female	40 (31.5%)	17 (25.0%)	23 (39.0%)	
Tumor location				
1. tonsil	91 (71.70%)	44 (64.70%)	47 (79.70%)	**0.010**
2. palate	10 (7.90%)	10 (14.70%)	0 (0.00%)	
3. root of the tongue	22 (17.30%)	13 (19.10%)	9 (15.30%)	
4.other oropharynx	4 (3.10%)	1 (1.50%)	3 (5.10%)	
Histopathological grading				
G1	7 (5.5%)	6 (8.8%)	1 (1.7%)	0.054
G2	55 (43.3%)	34 (50.0%)	21 (35.6%)	
G3	19 (15.0%)	7 (10.3%)	12 (20.3%)	
n.d.	46 (36.2%)	21 (30.9%)	25 (42.4%)	
Tumor depth (T)				
T1	13 (10.2%)	8 (11.8%)	5 (8.5%)	0.743
T2	42 (33.1%)	24 (35.3%)	18 (30.5%)	
T3	44 (34.6%)	22 (32.4%)	22 (37.3%)	
T4	27 (21.3%)	13 (19.1%)	14 (23.7%)	
Tx	1 (0.8%)	1 (1.5%)	0 (0.0%)	
Lymph node metastasis				
N 0–1	52 (40.9%)	36 (52.9%)	16 (27.1%)	**0.005**
N 2–3	74 (58.3%)	32 (47.1%)	42 (71.2%)	
Nx	1 (0.8%)		1 (1.7%)	
General treatment regimen				
Radiotherapy	31 (24.4%)	24 (35.3%)	7 (11.9%)	**0.003**
Radiochemotherapy	96 (75.6%)	44 (64.7%)	52 (88.1%)	

Values are presented as means ± standard deviations. n.d., not determined. Significant *p* values are marked in bold.

**Table 2 cancers-13-03256-t002:** Overall survival (OS) depending on pre-treatment parameters in HPV−/HPV+ patients: univariate and multivariate analysis.

Variable	OS HPV−				OS HPV+			
	Univariate Analysis		Multivariate Analysis		Univariate Analysis		Multivariate Analysis	
	HR (95% CI)	*p*-Value	HR (95% CI)	*p*-Value	HR (95% CI)	*p*-Value	HR (95% CI)	*p*-Value
Hb 0 [g/dL] >11.8 vs. <11.8	0.65 (0.15–2.79)	0.563			1.65 (0.21–12.86)	0.635		
RetHb 0 [/mm^3^] >34.0 vs. <34.0	0.9 (0.39–2.06)	0.803			2.42 (0.65–9.08)	0.190	4.33 (0.91–20.51)	0.065
RBC 0 [/mm^3^] >4.6 vs. <4.6	0.46 (0.18–1.13)	0.091			0.76 (0.25–2.37)	0.641		
Ret 0 [/mm^3^] >37.9 vs. <37.9	0.6 (0.26–1.38)	0.229			**0.22** **(0.06–0.82)**	**0.025**		
WBC 0 [/mm^3^] >8.33 vs. <8.33	2.33 (0.99–5.47)	0.053			**4.17** **(1.25–13.93)**	**0.020**		
CLC 0 [/mm^3^] >1.10 vs. <1.10	0.7 (0.28–1.74)	0.442			0.42 (0.11–1.56)	0.197		
CNC 0 [/mm^3^] >4.96 vs. <4.96	1.82 (0.78–4.22)	0.166			2.54 (0.75–8.52)	0.132		
CMC 0 [/mm^3^] >0.93 vs. <0.93	**2.41** **(1.01–5.74)**	**0.048**	**4.24** **(1.52–11.84)**	**0.006**	3.37 (0.72–15.72)	0.122	**17.8** **(1.89–167.47)**	**0.012**
PLT 0 [/mm^3^] >240 vs. <240	**2.53** **(1.07–5.99)**	**0.035**			1.1 (0.34–3.58)	0.872	0.29 (0.07–1.18)	0.085
NLR 0 >2.13 vs. <2.13	1.44 (0.65–3.18)	0.364	0.36 (0.12–1.10)	0.074	**4.76** **(1.29–17.57)**	**0.019**		
MLR 0 >0.43 vs. <0.43	**3.73** **(1.71–8.15)**	**0.001**	**3.83** **(1.46–10.02)**	**0.006**	0.47 (0.06–3.64)	0.470	0.07 (0.00–1.17)	0.064
PLR 0 >131.29 vs. <131.29	1.67 (0.76–3.69)	0.204	**3.6** **(1.15–11.31)**	**0.028**	1.89 (0.59–6.03)	0.281		
SII 0 >448.60 vs. <448.60	1.95 (0.81–4.68)	0.135			**5.67** **(1.24–25.94)**	**0.025**	**11.1** **(2.02–60.97)**	**0.006**
Treatment regimen RT/CRT	0.93 (0.42–2.06)	0.858			0.60 (0.08–4.64)	0.623		

0, before treatment; Hb, hemoglobin level; RetHb, ret-hemoglobin level; RBC, red blood cells; Ret, reticulocyte count; WBC, white blood cell count; CLC, circulating lymphocyte count; CNC, circulating neutrophil count; CMC, circulating monocyte count; PLT, platelet cell count. NLR, neutrophil/lymphocyte ratio; LMR, lymphocyte/monocyte ratio; PLR, platelet/lymphocyte ratio; SII, systemic immune inflammation index; RT, radiotherapy; CRT, chemoradiotherapy. Significant *p* values are marked in bold.

**Table 3 cancers-13-03256-t003:** Disease-free survival (DFS) depending on pre-treatment parameters in HPV−/HPV+ patients: univariate and multivariate analysis.

Variable	DFS HPV−				DFS HPV+			
	Univariate Analysis		Multivariate Analysis		Univariate Analysis		Multivariate Analysis	
	HR (95% CI)	*p*-Value	HR (95% CI)	*p*-Value	HR (95% CI)	*p*-Value	HR (95% CI)	*p*-Value
Hb 0 [g/dL] >13.5 vs. <13.5	**0.38** **(0.17–0.87)**	**0.021**			1.15 (0.27–4.82)	0.848	**10.44** **(1.31–83.38)**	**0.027**
RetHb 0 [/mm^3^] >36.4 vs. <36.4	1.47 (0.57–3.78)	0.423			1.30 (0.26–6.43)	0.750		
RBC 0 [/mm^3^] >4.6 vs. <4.6	**0.39** **(0.15–0.97)**	**0.043**	**0.21** **(0.07–0.66)**	**0.008**	0.43 (0.1–1.81)	0.252	**0.07** **(0.01–0.60)**	**0.015**
Ret 0 [/mm^3^] >54.7 vs. <54.7	0.64 (0.26–1.56)	0.323			0.23 (0.05–1.12)	0.068		
WBC 0 [/mm^3^] >8.33 vs. <8.33	1.57 (0.67–3.71)	0.301			2.42 (0.49–12.03)	0.280		
CLC 0 [/mm^3^] >1.28 vs. <1.28	0.61 (0.25–1.47)	0.271			**0.19** **(0.05–0.77)**	**0.020**	**0.17** **(0.03–0.97)**	**0.046**
CNC 0 [/mm_3_] >4.23 vs. <4.23	1.23 (0.56–2.72)	0.609			1.57 (0.38–6.6)	0.534		
CMC 0 [/mm^3^] >0.46 vs. <0.46	0.77 (0.32–1.85)	0.557			**0.16** **(0.03–0.8)**	**0.026**	0.20 (0.04–1.07)	0.060
PLT 0 [/mm^3^] >319 vs. <319	**2.77** **(1.18–6.52)**	**0.020**			1.59 (0.19–12.91)	0.667		
NLR 0 >2.29 vs. <2.29	1.63 (0.73–3.63)	0.232			**6.02** **(1.21–29.87)**	**0.028**		
MLR 0 >0.27 vs. <0.27	0.62 (0.28–1.38)	0.242	**0.20** **(0.06–0.62)**	**0.006**	0.46 (0.11–1.94)	0.291		
PLR 0 >173.20 vs. <173.20	**2.88** **(1.30–6.38)**	**0.009**	**2.74** **(1.02–7.41)**	**0.046**	1.77 (0.36–8.75)	0.487		
SII 0 >462.58 vs. <462.58	2.22 (0.92–5.35)	0.075	3.11 (0.90–10.77)	0.074	**8.48** **(1.04–68.98)**	**0.046**		
Treatment regimen	0.71 (0.31–1.64)	0.422			0.00	0.993		

0, before treatment; Hb, hemoglobin level; RetHb, ret-hemoglobin level; RBC, red blood cells; Ret, reticulocyte count; WBC, white blood cell count; CLC, circulating lymphocyte count; CNC, circulating neutrophil count; CMC, circulating monocyte count; PLT, platelet cell count. NLR, neutrophil/lymphocyte ratio; LMR, lymphocyte/monocyte ratio; PLR, platelet/lymphocyte ratio; SII, systemic immune inflammation index; RT, radiotherapy; CRT, chemoradiotherapy. Significant *p* values are marked in bold.

**Table 4 cancers-13-03256-t004:** Overall survival (OS) depending on post-treatment parameters in HPV−/HPV+ patients: univariate and multivariate analysis.

Variable	OS HPV−				OS HPV+			
	Univariate Analysis		Multivariate Analysis		Univariate Analysis		Multivariate Analysis	
	HR (95% CI)	*p*-Value	HR (95% CI)	*p*-Value	HR (95% CI)	*p*-Value	HR (95% CI)	*p*-Value
Hb 1 [g/dL] >10.1 vs. <10.1	**0.27** **(0.08–0.93)**	**0.037**			0.47 (0.06–3.77)	0.478		
RetHb 1 [/mm^3^] >37.8 vs. <37.8	1.37 (0.50–3.77)	0.537			0.67 (0.14–3.17)	0.613		
RBC 1 [/mm^3^] >3.75 vs. <3.75	0.53 (0.24–1.17)	0.118	**0.32** **(0.12–0.86)**	**0.024**	0.72 (0.23–2.29)	0.580		
Ret 1 [/mm^3^] >27.5 vs. <27.5	4.15 (0.56–30.94)	0.166	11.24 (0.99–127.94)	0.051	0.48 (0.14–1.65)	0.245		
WBC 1 [/mm^3^] >3.74 vs. <3.74	0.83 (0.36–1.93)	0.664	**0.16** **(0.03–0.93)**	**0.042**	2.60 (0.70–9.64)	0.152		
CLC 1 [/mm^3^] >0.83 vs. <0.83	0.90 (0.36–2.27)	0.823			**8.70** **(2.26–33.54)**	**0.002**	**11.37** **(2.61–49.64)**	**0.001**
CNC 1 [/mm^3^] >2.32 vs. <2.32	1.15 (0.43–3.06)	0.782	**7.48** **(1.00–56.06)**	**0.050**	1.51 (0.40–5.61)	0.541		
CMC 1 [/mm^3^] >0.42 vs. <0.42	1.70 (0.68–4.27)	0.260			0.94 (0.30–2.90)	0.908		
PLT 1 [/mm^3^] >313 vs. <313	2.06 (0.89–4.77)	0.092	**3.37** **(1.18–9.62)**	**0.023**	2.05 (0.26–16.14)	0.495		
NLR 1 >6.45 vs. <6.45	1.15 (0.48–2.79)	0.754			0.43 (0.12–1.60)	0.209		
MLR 1 >0.63 vs. <0.63	0.85 (0.38–1.91)	0.697	0.35 (0.12–1.03)	0.057	0.36 (0.09–1.39)	0.138		
PLR 1 >404.17 vs. <404.17	**2.55** **(1.13–5.77)**	**0.024**			0.87 (0.28–2.72)	0.815		
SII 1 >2763 vs. <2763	**3.54** **(1.28–9.80)**	**0.015**	**4.69** **(1.23–17.97)**	**0.024**	2.12 (0.46–9.81)	0.338	4.17 (0.76–22.88)	0.100

0, before treatment; Hb, hemoglobin level; RetHb, ret-hemoglobin level; RBC, red blood cells; Ret, reticulocyte count; WBC, white blood cell count; CLC, circulating lymphocyte count; CNC, circulating neutrophil count; CMC, circulating monocyte count; PLT, platelet cell count. NLR, neutrophil/lymphocyte ratio; LMR, lymphocyte/monocyte ratio; PLR, platelet/lymphocyte ratio; SII, systemic immune inflammation index. Significant *p* values are marked in bold.

**Table 5 cancers-13-03256-t005:** Disease-free survival (DFS) depending on post-treatment parameters in HPV−/HPV+ patients: univariate and multivariate analysis.

Variable	DFS HPV−				DFS HPV+			
	Univariate Analysis		Multivariate Analysis		Univariate Analysis		Multivariate Analysis	
	HR (95% CI)	*p*-Value	HR (95% CI)	*p*-Value	HR (95% CI)	*p*-Value	HR (95% CI)	*p*-Value
Hb 1 [g/dL] >10.8 vs. <10.8	**0.37** **(0.15–0.89)**	**0.027**			2.35 (0.29–19.11)	0.424		
RetHb 1 [/mm^3^] >38.0 vs. <38.0	2.28 (0.76–6.82)	0.142			0.72 (0.09–5.98)	0.761		
RBC 1 [/mm^3^] >3.75 vs. <3.75	**0.42** **(0.19–0.93)**	**0.033**	**0.26** **(0.11–0.66)**	**0.004**	1.56 (0.31–7.71)	0.589		
Ret 1 [/mm^3^] >27.5 vs. <27.5	4.03 (0.54–30.11)	0.174			0.39 (0.09–1.76)	0.221		
WBC 1 [/mm^3^] >3.21 vs. <3.21	0.57 (0.23–1.44)	0.233			17780630	0.994		
CLC 1 [/mm^3^] >0.37 vs. <0.37	0.56 (0.22–1.42)	0.226			4.03 (0.50–32.73)	0.193		
CNC 1 [/mm^3^] >2.69 vs. <2.69	0.98 (0.40–2.35)	0.956			1.66 (0.40–6.95)	0.488		
CMC 1 [/mm^3^] >0.47 vs. <0.47	0.50 (0.22–1.11)	0.087			1.31 (0.33–5.22)	0.706		
PLT 1 [/mm^3^] >261 vs. <261	1.07 (0.48–2.39)	0.873			**4.26** **(1.06–17.07)**	**0.041**	**7.97** **(1.55–41.00)**	**0.013**
NLR 1 >13.24 vs. <13.24	**3.16** **(1.18–8.50)**	**0.022**			1.17 (0.14–9.52)	0.882		
MLR 1 >0.63 vs. <0.63	0.66 (0.29–1.51)	0.325	0.40 (0.15–1.03)	0.058	0.48 (0.10–2.38)	0.368	0.21 (0.04–1.31)	0.095
PLR 1 >380 vs. <380	2.17 (0.95–4.99)	0.068	**3.96** **(1.46–10.77)**	**0.007**	0.80 (0.20–3.18)	0.747		
SII 1 >2730 vs. <2730	**3.25** **(1.27–8.28)**	**0.014**			1.33 (0.16–10.81)	0.790		

0, before treatment; Hb, hemoglobin level; RetHb, ret-hemoglobin level; RBC, red blood cells; Ret, reticulocyte count; WBC, white blood cell count; CLC, circulating lymphocyte count; CNC, circulating neutrophil count; CMC, circulating monocyte count; PLT, platelet cell count. NLR, neutrophil/lymphocyte ratio; LMR, lymphocyte/monocyte ratio; PLR, platelet/lymphocyte ratio; SII, systemic immune inflammation index. Significant *p* values are marked in bold.

**Table 6 cancers-13-03256-t006:** Pre-treatment and post-treatment prognostic factors for OS and DFS in HPV−/HPV+ patients.

Prognostic Factors	OS HPV−	DFS HPV−	OS HPV+	DFS HPV+
Poor prognostic factors in UVA (before treatment)	Higher CMC Higher PLT Higher MLR	Higher PLT Higher PLR Lower Hb Lower RBC	Higher WBC Higher NLR Higher SII Lower Ret	Higher NLR Higher SII Lower CLC Lower CMC
Poor prognostic factors in MVA (before treatment)	Higher CMC Higher MLR Higher PLR	Higher PLR Lower RBC Lower MLR	Higher CMC Higher SII	Higher Hb Lower RBC Lower CLC
Poor prognostic factors in UVA (after treatment)	Higher PLR Higher SII Lower Hb	Higher NLR Higher SII Lower Hb Lower RBC	Higher CLC	Higher PLT
Poor prognostic factors in MVA (after treatment)	Higher PLT Higher SII Lower RBC Lower WBC	Higher PLR Lower RBC	Higher CLC	Higher PLT

OS, overall survival; DFS, disease-free survival; UVA, univariate analysis; MVA, multivariate analysis; Hb, hemoglobin level; RetHb, ret-hemoglobin level; RBC, red blood cell count; WBC, white blood cell count; CLC, circulating lymphocyte count; CNC, circulating neutrophil count; CMC, circulating monocyte count; PLT, platelet cell count. NLR, neutrophil/lymphocyte ratio; LMR, lymphocyte/monocyte ratio; PLR, platelet/lymphocyte ratio; SII, systemic immune inflammation index.

## Data Availability

The data presented in this study are available in this article and Appendix A.

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
