# Peer review of "Comparison of Selected Immune and Hematological Parameters and Their Impact on Survival in Patients with HPV-Related and HPV-Unrelated Oropharyngeal Cancer"

_cancers, 2021, doi:10.3390/cancers13133256_

Round 1

Reviewer 1 Report

The authors have really made an effort to respond to my previous comments. Thank you!

Author Response

Dear Editors,

Dear Reviewers,

 Thank you for peer reviewing of our manuscript  cancers-1255922, entitled " Comparison of Selected Immune and Hematological Parameters and Their Impact on Survival in Patients with HPV-related and HPV- not Related Oropharyngeal Cancer".

Thank you for your questions and comments. We have fully addressed all the comments and my responses appear below. English has been improved. Our revised work includes corrections according to reviewers’ comments in the text. Our revisions, made according to reviewers’ comments, are marked using the „Track Changes” function. Supplementary materials are added.

We take this opportunity to express my gratitude to the reviewers for their constructive and useful remarks. Their comments allowed us to identify areas in my manuscript that needed modification.

We also thank you for allowing me to resubmit a revised copy of the manuscript.

We hope that the revised manuscript is now acceptable for publication in Cancers.

Yours sincerely,

Beata Jabłońska.

Responses to Reviewers' comments

Reviewer 1

Comment:

The authors have really made an effort to respond to my previous comments. Thank you!

Answer:

Thank you for appreciating our work. Thank you very much for your positive opinion.

Reviewer 2 Report

the authors have done a lot to improve the organization and description of their work.

Author Response

Dear Editors,

Dear Reviewers,

 Thank you for peer reviewing of our manuscript  cancers-1255922, entitled " Comparison of Selected Immune and Hematological Parameters and Their Impact on Survival in Patients with HPV-related and HPV- not Related Oropharyngeal Cancer".

Thank you for your questions and comments. We have fully addressed all the comments and my responses appear below. English has been improved. Our revised work includes corrections according to reviewers’ comments in the text. Our revisions, made according to reviewers’ comments, are marked using the „Track Changes” function. Supplementary materials are added.

We take this opportunity to express my gratitude to the reviewers for their constructive and useful remarks. Their comments allowed us to identify areas in my manuscript that needed modification.

We also thank you for allowing me to resubmit a revised copy of the manuscript.

We hope that the revised manuscript is now acceptable for publication in Cancers.

Yours sincerely,

Beata Jabłońska.

Responses to Reviewers’ comments

Reviewer 2

Comment:

the authors have done a lot to improve the organization and description of their work. This manuscript may benefit from a reorganization. 

Answer:

Thank you for appreciating our work. Thank you very much for your positive opinion.

Reviewer 3 Report

The authors did a good job in addressing most of the reviewer's comments. However, the novelty and significance of this study is not high.

Although the authors emphasized their study included more parameters than the study published by Huang et al. in 2015, in this study, they are only using the simple routine laboratory blood parameters, with different calculation methods.

In addition,  Huang's study included more than 700 patient cases, this study only includes 127 patients with HPV+ and HPV- OPC.   

Second, they concluded many different pre-treatment and post-treatment immune and hematological prognostic factors for OS and DFS in HPV+ and HPV- patients,  the clinical significance of using all these immune ratios as biomarkers is low, and the physicians would be less interested in this study.

Author Response

Dear Editors,

Dear Reviewers,

 Thank you for peer reviewing of our manuscript  cancers-1255922, entitled " Comparison of Selected Immune and Hematological Parameters and Their Impact on Survival in Patients with HPV-related and HPV- not Related Oropharyngeal Cancer".

Thank you for your questions and comments. We have fully addressed all the comments and my responses appear below. English has been improved. Our revised work includes corrections according to reviewers’ comments in the text. Our revisions, made according to reviewers’ comments, are marked using the „Track Changes” function. Supplementary materials are added.

We take this opportunity to express my gratitude to the reviewers for their constructive and useful remarks. Their comments allowed us to identify areas in my manuscript that needed modification.

We also thank you for allowing me to resubmit a revised copy of the manuscript.

We hope that the revised manuscript is now acceptable for publication in Cancers.

Yours sincerely,

Beata Jabłońska.

Responses to Reviewers’ comments

Reviewer 3

Comment:

The authors did a good job in addressing most of the reviewer's comments. However, the novelty and significance of this study is not high.

Answer:

Thank you for appreciating our work. There are only a few paper related to this subject and our study is a Polish / Central European voice in the worldwide discussion. Moreover, it may be used in a further meta-analysis. So far, there is no meta-analysis regarding the prognostic role of the hematological and immune parameters in HPV- and HPV+ OPC patients.

Comment

Although the authors emphasized their study included more parameters than the study published by Huang et al. in 2015, in this study, they are only using the simple routine laboratory blood parameters, with different calculation methods.

Answer

The cited study by Huang et al. is very good. Therefore, we have cited and described it, as well as compared to the our research. Our study is similar to the above mentioned research, because we have also used routine blood parameters. Certainly, there are also differences between these two papers, including a number of analyzed parameters and calculation methods. Our study is a supplementation of the worldwide literature regarding the prognostic role of hematological and immune parameters in HPV- and HPV+ patients.

Comment

In addition,  Huang's study included more than 700 patient cases, this study only includes 127 patients with HPV+ and HPV- OPC.   

Answer

Although our study includes a smaller cohort, it is a Polish / Central European voice in the discussion on this subject. Huang’s study involves population from Canada (North America), while our research represents European population, which completes the knowledge on the subject.

Comment

Second, they concluded many different pre-treatment and post-treatment immune and hematological prognostic factors for OS and DFS in HPV+ and HPV- patients,  the clinical significance of using all these immune ratios as biomarkers is low, and the physicians would be less interested in this study.

Answer

Post-treatment immune and hematological parameters have been inserted according to a Reviewer’s suggestion. The aim of our study was a comprehensive analysis of the prognostic significance of all routine blood parameters in HPV- and HPV+ OPC patients. In our opinion, because of their simplicity and availability, the phycisians should be interested in the use of tchem in the clinical practice.

This manuscript is a resubmission of an earlier submission. The following is a list of the peer review reports and author responses from that submission.

Round 1

Reviewer 1 Report

The Results section is poorly organized, not only short of descriptive text (1 page of text versus 21 pages of Tables/Figures), most table headers are not lined up correctly with corresponding contents. 

This made the reviewer very difficult to evaluate the merit of these results. 

Furthermore, could you please extract critical results into the main text, elaborate them in more detail, and move less important information to the Appendix or Supplementary sections. 

In 2.2. Study design, please add information regarding how the basic laboratory blood parameters were measured, including on what equipments or reagent brand etc. 

Author Response

Dear Editor,

Dear Reviewer,

 Thank you for peer reviewing of our manuscript  cancers-1189091, entitled " Comparison of Selected Immune and Hematological Parameters and Their Impact on Survival in Patients with HPV-related and HPV- not Related Oropharyngeal Cancer".

Thank you for your questions and comments. We have fully addressed all the comments and my responses appear below. Our revised work includes corrections according to reviewers’ comments in the text. The changes, made according to reviewers’ comments, are highlighted in red print in the text. Supplementary materials are added.

We take this opportunity to express my gratitude to the reviewers for their constructive and useful remarks. Their comments allowed us to identify areas in my manuscript that needed modification.

We also thank you for allowing me to resubmit a revised copy of the manuscript.

We hope that the revised manuscript is now acceptable for publication in Cancers.

Yours sincerely,

Beata Jabłońska.

Responses to Reviewers

Reviewer 1

The Results section is poorly organized, not only short of descriptive text (1 page of text versus 21 pages of Tables/Figures), most table headers are not lined up correctly with corresponding contents. 

This made the reviewer very difficult to evaluate the merit of these results. 

Answer: The result section has been improved according to the Reviewer’s suggestions. The descriptive text has been longed and tables have been corrected as follows:

3.1. General characteristics

The general clinical characteristic of 127 patients is presented in Table 1.

The both groups were comparable regarding the age and gender structure. The mean age was 60.85±7.48 (37-79) and 60.36±9.67 (30-80) years in the HPV- and HPV+ groups, respectively (p=0.745). The male gender was predominant in the both groups. There was 51 (75.1%) and 36 (61.0%) males in the HPV- and HPV+ groups, respectively (p=0.133).

The tonsil was the most common OPC location in the both groups, but the incidence of the tumor location was different depending the HPV status. The tonsil location was the most frequent (47 (79.7%)) in HPV+ patients, and in this location was noted only in 44 (64.7%) HPV- patients (p=0.010). The palate location was not observed in HPV+ patients, and this location was noted in 10 (14.7%) HPV- patients.                    

Concerning the histopathological findings, in the both groups, G2 grading and T2/T3 staging were the most common. G2 grading was reported in 34 (72.3%) and 21 (61.8%) patients in HPV- and HPV+ patients, respectively. G3 grading was more frequently noted in HPV+ patients compared to HPV- ones (14.9% vs. 35.3%; p=0.049). The tumor depth was similar in the both groups. It should be emphasized that the HPV+ patients had the more advanced nodal status compared to HPV- patients ( 71.2% vs. 47.1% N2-3; p=0.003)

The significantly higher regional advancement of the HPV+ tumors was associated with the difference in the treatment regimen in the both groups. RCT was significantly more frequently used in the HPV+ patients compared to HPV- patients (88.1% vs. 64.7%; p=0.003). 

3.2. Laboratory results before and after treatment

The basic laboratory results in all patients and in both HPV- and HPV+ groups before and after treatment are presented and compared in Table S1.

Most pre-treatment (0) and post-treatment (1) laboratory results were comparable in both groups. Moreover, a decrease of the most parameters following RT/CRT was noted in our study. Only Ret hemoglobin (RetHb) has increased after the treatment in HPV+ patients. Therefore, a significantly higher RetHb1 level was noted in HPV+ patients (35.01±1.43) compared to those HPV- (33.50±3.38). A reticulocyte count (Ret) before (51.97±24.96 vs. 60.56±22.73, p=0.052) and after treatment (51.20±25.29 vs 48.62±23.88; p=0.574),was comparable in HPV- and HPV+ groups, respectively), but the Ret decrease was significantly greater in HPV+ patients compared to HPV- ones (0.89±29.78 vs. 12.30±30.51, p=0.044). A lower WBC 0 was reported in HPV+ patients compared to HPV- participants (6.45±1.91 vs. 7.15±2.03, p=0.048), while the difference in WBC1 between both groups was not statistically significant (4.35±2.25 vs. 5.12±2.29, p=0.059). The WBC decrease was observed in the both groups. CLC1 (p=0.004) and CMC1 (p=0.012) were significantly lower in HPV+ patients, while these pre-treatment parameters were comparable in HPV- and HPV+ groups (p=0.842 for CLC0, p=0.057 for CMC0). A significantly lower PLT1 count was recorded in HPV+ patients compared to HPV- ones (208.15±65.18 vs. 250.31±118.11, p=0.016), while PLT0 was similar in HPV+ and HPV- groups (236.76±57.94 vs. 256.93±77.16, p=0.103). It was associated with a greater PLT decrease in HPV+ patients (p=0.218).

The NLR 0/1, MLR 0/1, PLR 0/1 and SII 0/1 were comparable in both groups. The results were presented in Table S2.

 3.2. Comparison of clinical and pathological characteristics depending on the values of immune ratios in HPV- and HPV+ patients

The differences between low and high pre-treatment immune ratios groups (NLR 0/1, MLR 0/1, PLR 0/1 and SII 0/1) were analyzed.

In the low NLR HPV- subgroup, the significantly greater incidence of tonsil location (34 (73.9%)) compared to the incidence (10 (45.5%)) in the high NLR subgroup (p=0.039) was reported. In HPV+ patients, the incidence of tonsil location was comparable in both NLR subgroups (30 (81.1%) vs. 17 (77.3%), respectively (p=0.748). There was no significant difference in terms of the other clinicopathological parameters between two NLR groups regardless of HPV status (Table S3).

In the low MLR HPV- subgroup, the highest G3 grading was significantly more frequent (7 (25.0%)) compared to the high MLR subgroup (0 (0.0%)) (p=0.026). In comparison of the determined grading, G2 was the commonest grading in both subgroups. In HPV+ patients, histological grading was comparable in both MLR subgroups. The other clinicopathological parameters were similar in both MLR subgroups (Table S4).

In the low PLR HPV+ subgroup, significantly more frequent smoking was observed compared to the high PLR subgroup: 17 (43.6%) vs. 3 (15.0%) (p= 0.042). This difference was not noted in HPV- patients. The other clinicopathological parameters were comparable in both PLR subgroups (Table S5).

In HPV- patients, a higher initial BMI was noted in patients with the low SII compared to patients with the high SII: 26.6±4.8 vs. 24.1±4.1 kg/m2, respectively (p=0.028). In HPV+ patients, G3 grading was more frequent in the low SII group compared to patients with the high SII: 11 (47.8%) vs. 1 (9.1%) (p=0.021). Moreover, significantly more frequent smoking was seen in the low SII HPV+ subgroup compared to patients with the high SII: 17 (43.6%) vs. 3 (15.0%) (p=0.042) (Table S6). All comparisons are presented in Tables S3-S6.

                                                                       3.3. Overall survival and disease-free survival depending on HPV status

OS and DFS in both HPV groups are presented in Tables S7-S10 and Figure 2, as well as Figure S1 A,B,C, and Figure S2 A,B,C. OS and DFS depending on HPV status was assessed using Cox regression univariate (UVA) and multivariate analysis (MVA). The prognostic factors determined in UVA were confirmed and presented using Kaplan-Meier curves. Generally, HPV status was a very strong prognostic factor for OS and DFS in our patients. OS and DFS were significantly better in HPV+ patients compared to HPV- ones (p = 0.0008 and p = 0.0009, respectively) (Figure 2).

Figure 2. Comparison of overall survival (OS) and disease-free survival (DFS) between HPV- and HPV+ groups.

3.3.1. Overall survival depending on pre-treatment parameters

3.3.1.1.HPV-

In UVA, poor prognostic factors for OS in HPV- patients were as follows: a higher CMC (HR 2.41, 95%CI 1.01-5.74, p = 0.048), a higher PLT (HR 2.53, 95%CI 1.07-5.99, p = 0.035), and a higher MLR (HR 3.73, 95%CI 1.71-8.15, p = 0.001). A higher WBC (HR 2.33, 95%CI 0.99-5.44, p = 0.053) marginally predicted inferior OS in HPV-patients.

In MVA, a higher CMC (HR 4.24, 95%CI 1.52-11.84, p = 0.006), a higher MLR (HR 3.83, 95%CI 1.46-10.02, p = 0.006), and a higher PLR (HR 3.60, 95%CI 1.15-11.31, p = 0.028) were poor prognostic factors for OS.

3.3.1.2.HPV+

In UVA, poor prognostic factors for OS in HPV+ patients were as follows: a higher WBC (HR 4.17, 95%CI 1.25-13.93, p = 0.020), a higher NLR (HR 4.76, 95%CI 1.29-17.57, p = 0.019), and a higher SII (HR 5.67, 95%CI 1.24-25.94, p = 0.025). A higher Ret was associated with higher OS (HR 0.22, 95%CI 0.06-0.82, p = 0.025),

In MVA, a higher CMC (HR 17.18, 95%CI 1.89-167.47, p = 0.012), and a higher SII (HR 11.10, 95%CI 2.02-60.97, p = 0.006) were poor prognostic factors for OS.

3.3.2. Disease-free survival depending on pre-treatment parameters

3.3.2.1.HPV-

In UVA, poor prognostic factors for DFS in HPV- patients were as follows: a higher PLT (HR 2.77, 95%CI 1.18-6.52, p = 0.020), and higher PLR (HR 2.88, 95%CI 1.30-6.38, p = 0.009), whereas a higher RBC (HR 0.39, 95%CI 0.15-0.97, p = 0.043) predicted better DFS.

In MVA, a higher PLR (HR 2.74, 95%CI 1.02-7.41, p = 0.046) was a poor prognostic factor for DFS, whereas a higher RBC (HR 0.21, 95%CI 0.07-0.66, p = 0.008), a higher MLR (HR 0.20, 95%CI 0.06-0.62, p = 0.006) were associated with higher DFS.

3.3.2.2.HPV+

In UVA, poor prognostic factors for OS in HPV-+ patients were as follows: a higher NLR (HR 6.02, 95%CI 1.21-29.87, p = 0.028), and a higher SII (HR 8.48, 95%CI 1.04-68.98, p = 0.046), whereas a higher CLC (HR 0.19, 95%CI 0.05-0.77, p = 0.020), a higher CMC (HR 0.16, 95%CI 0.03-10.8, p = 0.026) predicted higher DFS.

In MVA, higher Hb (HR 10.44, 95%CI 1.31-38.83, p = 0.027) was poor prognostic factors for DFS, whereas a higher CLC (HR 0.17, 95%CI 0.03-0.97, p = 0.046) was associated with higher DFS. A higher CMC (HR 0.20, 95%CI 0.04-1.07, p = 0.060) marginally predicted higher DFS in HPV+ patients.

3.3.3. Overall survival depending on post-treatment parameters

3.3.3.1.HPV-

In UVA, poor prognostic factors for OS in HPV- patients were as follows: a higher PLR (HR 2.55, 95%CI 1.13-5.77, p = 0.024), and a higher SII (HR 3.54, 95%CI 1.28-9.80, p = 0.015). A higher Hb (HR 0.27, 95%CI 0.08-0.93, p = 0.037) predicted better OS in HPV-patients.

In MVA, a a higher PLT (HR 3.37, 95%CI 1.18-9.62, p = 0.023) and a higher SII (HR 4.69, 95%CI 1.23-17.97, p=0.024) were poor prognostic factors for OS. A higher RBC (HR 0.32, 95%CI 0.12-0.86, p=0.024), a higher WBC (HR 0.16, 95%CI 0.03-0.93, p=0.042) were associated with better OS.

3.3.3.2.HPV+

In UVA, only a higher CLC (HR 8.70, 95%CI 2.26-33.54, p = 0.002) was a poor prognostic factor for OS in HPV+ patients.

In MVA, also a higher CLC (HR 11.37, 95%CI 2.61-49.64, p = 0.001) was a poor prognostic factor for OS in HPV+ patients..

3.3.4. Disease-free survival depending on post-treatment parameters

3.3.4.1.HPV-

In UVA, poor prognostic factors for DFS in HPV- patients were as follows: a higher NLR (HR 3.16, 95%CI 1.18-8.50, p = 0.022) and a higher SII (HR 3.25, 95%CI 1.27-8.28, p = 0.014), whereas a higher Hb (HR 0.37, 95%CI 0.15-0.89, p = 0.027) and a higher RBC (HR 0.42, 95%CI 0.19-0.93, p = 0.033) were associated with better DFS.

In MVA, a higher PLR (HR 3.96, 1.46-10.77, p = 0.007) was a poor prognostic factor for DFS, whereas a higher RBC (HR 0.26, 95%CI 0.11-0.66, p = 0.004) was associated with higher DFS.

3.3.4.2.HPV+

In UVA, only a higher PLT (HR 4.26, 95%CI 1.06-17.07, p = 0.041) was a poor prognostic factors for OS in HPV-+ patients.

In MVA, also a higher PLT (HR 7.97, 95%CI 1.55-41.00, p = 0.013) was the only poor prognostic factor for DFS.

3.3.5. Summary of the analysis of OS and DFS in HPV- and HPV+ patients

In summary, our study showed different pre-treatment and post-treatment parameters predicting OS and DFS in HPV- and HPV+ patients. HPV status was the strongest predictor for OS and DFS. Generally, there were more immune and hematological parameters predicting survival in HPV- patients compared to HPV+ participants. In HPV+ patients, a high pre-treatment WBC, NLR, and SII significantly correlated with reduced OS, whereas high NLR SII significantly correlated with reduced DFS. A higher pre-treatment NLR and SII were significant poor prognostic factors for both OS and DFS in HPV+ group. These associations were not apparent in HPV- patients.

Thus, there are different pre-treatment and post-treatment immune and hematological prognostic factors for OS and DFS in HPV+ and HPV- patients.

Summary of differences regarding the impact of immune and hematological pre-treatment and post-treatment parameters on OS and DFS depending on HPV status was presented in Table 2.

Table 2. Pre-treatment and post-treatment prognostic factors for OS and DFS in HPV-/HPV+ patients.

Prognostic factors

OS HPV-

DFS HPV-

OS HPV+

DFS HPV+

Poor prognostic factors in UVA

(before treatment)

Higher CMC

Higher PLT

Higher MLR

Higher PLT

Higher PLR

Lower RBC

Higher WBC

Higher NLR

Higher SII

Lower Ret

Higher NLR

Higher SII

Lower CLC

Lower CMC

Poor prognostic factors in MVA

(before treatment)

Higher CMC

Higher MLR

Higher PLR

Higher PLT

Higher PLR

Lower RBC

Higher CMC

Higher SII

Higher Hb

Lower CLC

Poor prognostic factors in UVA

(after treatment)

Higher PLR

Higher SII

Lower Hb

Higher NLR

Higher SII

Lower Hb

Lower RBC

Higher CLC

Higher PLT

Poor prognostic factors in MVA

(after treatment)

Higher PLT

Higher SII

Lower RBC

Lower WBC

Higher PLR

Lower RBC

Higher CLC

Higher PLT

OS, overall survival, DFS, disease-free-survival; UVA, univariate analysis; MVA, multivariate analysis; Hb, hemoglobin level; RetHb, RBC, red blood cells; WBC, white blood cells count; CLC, circulating lymphocyte count; CNC, circulating neutrophil count; CMC, circulating monocyte count; PLT, platelet cells count.

NLR, neutrophil/lymphocyte ratio; LMR, lymphocyte/monocyte ratio; PLR, platelet/lymphocyte ratio; SII, systemic immune inflammation index.

Furthermore, could you please extract critical results into the main text, elaborate them in more detail, and move less important information to the Appendix or Supplementary sections. 

Answer: Most tables and figures have been located in the Supplementary. Only Table 1 (General characteristics) and Table 2 (Summary of the results regarding OS and DFS) as well as Figure 1 (flowchart diagram of our study) and Figure 2 (comparision of OS and DFS depending on HPV status) have been presented in the main text.

In 2.2. Study design, please add information regarding how the basic laboratory blood parameters were measured, including on what equipments or reagent brand etc. 

Answer: This information has been added as follows:

2.2.2. Laboratory blood investigations and analysis

The blood was obtained in standard conditions, the patients in fasting state, between 7:00 and 9:00, by means of a vacuum Becton Dickinson (Franklin Lakes, NJ, USA) system, to sample tubes with anticoagulant EDTA. The full blood count was determined using the Sysmex XN-2000 analyzer (Sysmex, Kobe, Japan). 

2.2.3. Confirmation of the HPV etiology

Confirmation of the HPV etiology was obtained from tissue material and/or extracellular circulating HPV DNA.

Tissue material

Formalin-fixed paraffin-embedded tumor samples were examined for high-risk HPV (HR-HPV) infection using double-check algorithm including immunohistochemical assessment of P16(INK4A) protein expression followed by detection of HR-HPV DNA in tumor tissue using real time PCR. Only cases with both p16INK4A expression and HR-HPV DNA amplification were classified as truly HR HPV-positive [7].

Analysis of cfHPV16 DNA in plasma.

Peripheral blood (12 ml) was collected into K3EDTA tubes (Becton–Dickinson, New Jersey, Franklin Lakes, USA). Plasma was separated within an hour by double centrif-ugation at 300×g and 1000×g, both at 4 °C for 10 min. DNA was extracted (according to the manufacturer’s instructions) from 1 ml of plasma by the Genomic Mini AX Body Fluids kit (A&A Biotechnology, Gdynia, Poland). Each measurement consisted of a standard curve of three dilutions of plasmid construct containing HPV16 genome, neg-ative control and a samples. For HPV16 detection, reaction was performed using pri-mers and probe set for HPV16 genome. PCR reactions were performed using the Bio-Rad CFX96 qPCR instrument (Bio-Rad Laboratories, Hemel Hempstead, United Kingdom). If HPV16 was found, it’s presence would be confirmed with a second inde-pendent DNA isolation.

Reviewer 2 Report

I am sure that the authors have generated some interesting data; however, the presentation of the data in multi-segment tables is very distracting.

This manuscript may benefit from a reorganization. 

Also, if at all possible a schematic or cartoon figure outlining what the authors believe is happening may help get their point across.

The font size in the Tables needs to be reduced so that the appropriate data fits on a single line and does not require wrap-around text.

I would bold all the significant values in the tables so that it is easy to find the significant results.

I would request that many of the tables be moved to supplemental data.  They make it very difficult to read the manuscript in its present form. Table 1 is important for the main text but the rest of the tables could be moved to supplemental.

A summary table comparing the major differences between HPV+ and HPV- and their associated hematological differences would help with the organization of the paper.

Author Response

Dear Editor,

Dear Reviewer,

 Thank you for peer reviewing of our manuscript  cancers-1189091, entitled " Comparison of Selected Immune and Hematological Parameters and Their Impact on Survival in Patients with HPV-related and HPV- not Related Oropharyngeal Cancer".

Thank you for your questions and comments. We have fully addressed all the comments and my responses appear below. Our revised work includes corrections according to reviewers’ comments in the text. The changes, made according to reviewers’ comments, are highlighted in red print in the text. Supplementary materials are added.

We take this opportunity to express my gratitude to the reviewers for their constructive and useful remarks. Their comments allowed us to identify areas in my manuscript that needed modification.

We also thank you for allowing me to resubmit a revised copy of the manuscript.

We hope that the revised manuscript is now acceptable for publication in Cancers.

Yours sincerely,

Beata Jabłońska.

Responses to Reviewers

Reviewer 2

I am sure that the authors have generated some interesting data; however, the presentation of the data in multi-segment tables is very distracting.

This manuscript may benefit from a reorganization. 

Answer: The manuscript (including the main text, tables, and figures) has been reorganized. Please see details in the main text and supplementary.

Also, if at all possible a schematic or cartoon figure outlining what the authors believe is happening may help get their point across.

Answer: The figure as flowchart diagram has been added as follows:

Figure 1. Flowchart diagram for the study.

The font size in the Tables needs to be reduced so that the appropriate data fits on a single line and does not require wrap-around text.

Answer: All the tables have been reconstructed.

I would bold all the significant values in the tables so that it is easy to find the significant results.

Answer: The significant values have been bold in the tables (Please see tables in the supplementary materials).

I would request that many of the tables be moved to supplemental data.  They make it very difficult to read the manuscript in its present form. Table 1 is important for the main text but the rest of the tables could be moved to supplemental.

Answer: Most tables and figures have been located in the Supplementary. Only Table 1 (General characteristics) and Figure 1 (comparision of OS and DFS depending on HPV status) have been presented in the main text.

A summary table comparing the major differences between HPV+ and HPV- and their associated hematological differences would help with the organization of the paper.

Answer: It has been added as follows:

Table 2. Pre-treatment and post-treatment prognostic factors for OS and DFS in HPV-/HPV+ patients.

Prognostic factors

OS HPV-

DFS HPV-

OS HPV+

DFS HPV+

Poor prognostic factors in UVA

(before treatment)

Higher CMC

Higher PLT

Higher MLR

Higher PLT

Higher PLR

Lower RBC

Higher WBC

Higher NLR

Higher SII

Lower Ret

Higher NLR

Higher SII

Lower CLC

Lower CMC

Poor prognostic factors in MVA

(before treatment)

Higher CMC

Higher MLR

Higher PLR

Higher PLT

Higher PLR

Lower RBC

Higher CMC

Higher SII

Higher Hb

Lower CLC

Poor prognostic factors in UVA

(after treatment)

Higher PLR

Higher SII

Lower Hb

Higher NLR

Higher SII

Lower Hb

Lower RBC

Higher CLC

Higher PLT

Poor prognostic factors in MVA

(after treatment)

Higher PLT

Higher SII

Lower RBC

Lower WBC

Higher PLR

Lower RBC

Higher CLC

Higher PLT

OS, overall survival, DFS, disease-free-survival; UVA, univariate analysis; MVA, multivariate analysis; Hb, hemoglobin level; RetHb, RBC, red blood cells; WBC, white blood cells count; CLC, circulating lymphocyte count; CNC, circulating neutrophil count; CMC, circulating monocyte count; PLT, platelet cells count.

NLR, neutrophil/lymphocyte ratio; LMR, lymphocyte/monocyte ratio; PLR, platelet/lymphocyte ratio; SII, systemic immune inflammation index.

Reviewer 3 Report

In this study, the authors analyzed selected immune and hematological parameters of patients with HPV+ and HPV- oropharyngeal cancer, before and after radiotherapy / chemoradiotherapy (RT/CRT), and assessed the impact of these parameters on OS and DFS. All these peripheral blood inflammation factors impact on prognosis have been well studied in HNSCC, so the novelty of this study is very limited. In addition, the following concerns need to be addressed.

Major concerns:

  1. Remake Table1, Table2, Table3, they are all in mess orders, and note clearly the p value from which two groups’ comparison, “Values are presented as means and standard deviations” change to means ± standard deviations .

  1. The patients inclusion criteria was too simple, did the patients had surgery before radical RT or CRT?  

  1. The analysis included 127 adults, please explain why in the Table1 general patients clinicopathological characteristics, only 80 patients has Histopathological grading, the same question in all the Table 4 A-H.

  1. This study analysis designed based on HPV status, was HPV positive determined by p16 (HPV) status or contains low-risk stains, did the authors analyze the difference?

  1. In Table 5, 6, 7, the authors did the comparison between HPV- and HPV+ patients in OS and DFS, all depending on the analyzed parameters before treatment, did they compare these post-treatment parameters on patients OS and DFS?

  1. This study concludes that OS and DFS were significantly better in HPV+ patients compared to HPV- ones, there was no possible explanations in the discussion.

Author Response

Dear Editor,

Dear Reviewer,

 Thank you for peer reviewing of our manuscript  cancers-1189091, entitled " Comparison of Selected Immune and Hematological Parameters and Their Impact on Survival in Patients with HPV-related and HPV- not Related Oropharyngeal Cancer".

Thank you for your questions and comments. We have fully addressed all the comments and my responses appear below. Our revised work includes corrections according to reviewers’ comments in the text. The changes, made according to reviewers’ comments, are highlighted in red print in the text. Supplementary materials are added.

We take this opportunity to express my gratitude to the reviewers for their constructive and useful remarks. Their comments allowed us to identify areas in my manuscript that needed modification.

We also thank you for allowing me to resubmit a revised copy of the manuscript.

We hope that the revised manuscript is now acceptable for publication in Cancers.

Yours sincerely,

Beata Jabłońska.

Responses to Reviewers

Reviewer 3

In this study, the authors analyzed selected immune and hematological parameters of patients with HPV+ and HPV- oropharyngeal cancer, before and after radiotherapy / chemoradiotherapy (RT/CRT), and assessed the impact of these parameters on OS and DFS. All these peripheral blood inflammation factors impact on prognosis have been well studied in HNSCC, so the novelty of this study is very limited. In addition, the following concerns need to be addressed.

Major concerns:

  1. Remake Table1, Table2, Table3, they are all in mess orders, and note clearly the p value from which two groups’ comparison, “Values are presented as means and standard deviations” change to means ± standard deviations .

 Answer: Table 1,2,3 have been corrected as follows:

                                                                       Table 1. The general patients clinicopathological characteristics.

All n=127

HPV(-) n=68

HPV(+) n=59

p

Demographic characteristics

Age

60.62±8.54
(30-80)

60.85±7.48
(37-79)

60.36±9.67 (30-80)

0.745

Male

87(68.5%)

51 (75.1%)

36 (61.0%)

0.133

Female

40 (31.5%)

17 (25.0%)

23 (39.0%)

Tumor location

1. tonsil

91 (71.70%)

44 (64.70%)

47 (79.70%)

0.010

2. palate

10 (7.90%)

10 (14.70%)

0 (0.00%)

3. root of the tongue

22 (17.30%)

13 (19.10%)

9 (15.30%)

4.other oropharynx

4 (3.10%)

1 (1.50%)

3 (5.10%)

Histopathological grading

G1

7 (5.5%) 

6 (8.8%)

1 (1.7%)

0.054

G2

55 (43.3%)

34 (50.0%)

21 (35.6%)

G3

19 (15.0%)

7 (10.3%)

12 (20.3%)

n.d.

46 (36.2%)

21 (30.9%)

25 (42.4%)

Tumor depth (T)

T1

13 (10.2%)

8 (11.8%)

5 (8.5%)

0.743

T2

42 (33.1%)

24 (35.3%)

18 (30.5%)

T3

44 (34.6%)

22 (32.4%)

22 (37.3%)

T4

27 (21.3%)

13 (19.1%)

14 (23.7%)

Tx

1 (0.8%)

1 (1.5%)

0 (0.0%)

Lymph node metastasis

N 0-1

52 (40.9%)

36 (52.9%)

16 (27.1%)

0.005

N 2-3

74 (58.3%)

32 (47.1%)

42 (71.2%)

Nx

1 (0.8%)

1 (1.7%)

General treatment regimen

Radiotherapy

31 (24.4%)

24 (35.3%)

7 (11.9%)

0.003

Radiochemotherapy

96 (75.6%)

44 (64.7%)

52 (88.1%)

Values are presented as means and standard deviations.

n.d., not determined.

Table S1. Pre-treatment and post-treatment laboratory (peripheral blood morphology parameters) results.

HPV(-)

HPV(+)

p

Hb 0 [g/dl]

Hb 1 [g/dl]

Hb 01 [g/dl]

14.05±1.51

12.03±1.54

1.75±1.46                           

13.88±1.51

12.03±1.54

1.85±1.58

0.530

0.331

0.717

RetHb 0 [/mm3]  

RetHb 1 [/mm3]  

RetHb 01 [/mm3

34.04±3.36

33.50±3.38

0.48±4.31             

34.21±2.61

35.01±1.43

-0.73±2.91            

0.762

0.031

0.078

RBC 0 [/mm3]

RBC 1 [/mm3]

RBC 01 [/mm3]

4.51±0.53

4.00±0.55

0.51±0.52             

4.61±0.54

3.94±0.57

0.68±0.54

0.274

0.531

0.085

Ret 0 [/mm3]       

Ret 1 [/mm3]       

Ret 01 [/mm3]      

51.97±24.96

51.20±25.29

0.89±29.78                         

60.56±22.73         

48.62±23.88

12.30±30.51         

0.052

0.574

0.044

WBC 0 [/mm3]    

WBC 1 [/mm3]    

WBC 01 [/mm3]  

7.15±2.03

5.12±2.29

2.00±2.87

6.45±1.91

4.35±2.25

2.10±2.70

0.048

0.059

0.846

CLC 0 [/mm3]     

CLC 1 [/mm3]     

CLC 01 [/mm3]   

1.91±0.71

0.71±0.44

1.20±0.72                           

1.89±0.73

0.52±0.24

1.37±0.69

0.842

0.004

0.174

CNCC 0 [/mm3]  

CNCC 1 [/mm3]  

CNCC 01 [/mm3]

4.34±1.86

3.67±1.84

0.66±2.55             

3.77±1.49             

3.24±2.10

0.54±2.20

0.064

0.226

0.779

CMC 0 [/mm3]    

CMC 1 [/mm3]    

CMC 1 [/mm3]    

0.64±0.25             

0.61±0.32

0.04±0.33

0.56±0.24

0.48±0.20

0.08±0.28

0.057

0.012

0.502

PLT 0 [/mm3]

PLT 1 [/mm3]

PLT 01 [/mm3]

256.93±77.16       

250.31±118.11

7.25±119.63         

236.76±57.94

208.15±65.18       

28.61±60.37         

0.103

0.016

0.218

Values are presented as means ± standard deviations.

0, before treatment, 1, after treatment; 01, difference; Hb, hemoglobin level; RetHb, ret-hemoglobin level; RBC, red blood cells; RetC, reticulocyte count; WBC, white blood cells count; CLC, circulating lymphocyte count; CNC, circulating neutrophil count; CMC, circulating monocyte count; PLT, platelet cells count.

Table S2. Pre-treatment and post-treatment immune ratios.

HPV(-)

HPV(+)

p

NLR 0   

NLR 1   

NLR 01 

2.71±2.14

6.66±4.95

-3.94±5.28

2.31±1.35

7.59±6.38

-5.29±6.05

0.216

0.355

0.186

MLR 0   

MLR 1   

MLR 01 

0.36±0.16

1.00±0.54

-0.64±0.53

0.31±0.15

1.10±0.70

-0.79±0.65

0.053

0.385

0.154

PLR 0    

PLR 1    

PLR 01  

152.25±73.47

436.38±249.77

-283.25±228.41    

145.09±77.14

486.40±251.00      

-341,31±226.87    

0.593

0.265

0.156

SII 0

SII 1

SII 01

707.36±586.07      

1638.77±1406.1    

-929.62±1430.54                 

553.31±369.82

1471.13±1053.4

-917.81±948.99                  

0.084

0.453

0.957

Values are presented as means ± standard deviations.

0, before treatment, 1 after treatment; 01, difference; NLR, neutrophil/lymphocyte ratio (NLR); LMR, lymphocyte/monocyte ratio; PLR, platelet/lymphocyte ratio; SII, systemic immune inflammation index.

  1. The patients inclusion criteria was too simple, did the patients had surgery before radical RT or CRT?  

 Answer: The patients had not got surgery before radical RT or CRT. Inclusion and exclusion criteria have been extended as follows:

  1. The analysis included 127 adults, please explain why in the Table1 general patients clinicopathological characteristics, only 80 patients has Histopathological grading, the same question in all the Table 4 A-H.

Answer: For some cases, grading has not been established due to the scant available biopsy material. The number of such cases was evenly distributed among all groups of patients, which did not affect the obtained results. Moreover, grading in the case of OPC has little prognostic or predictive significance. Finally, information on the number of the not-determined cases and their percentage in all examined groups was added to Table 1.

  1. This study analysis designed based on HPV status, was HPV positive determined by p16 (HPV) status or contains low-risk stains, did the authors analyze the difference?

Answer: Tumor tissue samples were examined for high-risk HPV (HR-HPV) infection using double-check algorithm including immunohistochemical assessment of P16(INK4A) protein expression followed by detection of HR-HPV DNA in tumor tissue using real time PCR. Only cases with both p16INK4A expression and HR-HPV DNA amplification were classified as truly HR HPV-positive.

A detailed HPV status assessment protocol has been described previously. Information on the HPV status assessment methodology and a reference to the source article have been added in the “Material and methods” section of the revised manuscript.

Because we are now focusing on the patients with oropharyngeal cancer alone and p16 status is the most widely used surrogate marker for HPV associated oropharyngeal cancer, this paper would be strengthened by looking at the relationship between p16 expression in these patients and the other parameters (tumor viral load, circulating HPV16 DNA) measured in this study.

For the analysis of HPV DNA in the paraffin embedded biopsies patients were asked to bring them in the clinic/laboratory. After DNA isolation, the samples (biopsies) were returned to the patients, so at this moment we do not have samples to do IHC of p16.

  1. In Table 5, 6, 7, the authors did the comparison between HPV- and HPV+ patients in OS and DFS, all depending on the analyzed parameters before treatment, did they compare these post-treatment parameters on patients OS and DFS?

Answer: The impact of post-treatment parameters on patients OS and DFS has been additionally analyzed and these results have been added in Table 2 as well as Table S9 and Table S10.

 Table 2. Pre-treatment and post-treatment prognostic factors for OS and DFS in HPV-/HPV+ patients.

Prognostic factors

OS HPV-

DFS HPV-

OS HPV+

DFS HPV+

Poor prognostic factors in UVA

(before treatment)

Higher CMC

Higher PLT

Higher MLR

Higher PLT

Higher PLR

Lower RBC

Higher WBC

Higher NLR

Higher SII

Lower Ret

Higher NLR

Higher SII

Lower CLC

Lower CMC

Poor prognostic factors in MVA

(before treatment)

Higher CMC

Higher MLR

Higher PLR

Higher PLT

Higher PLR

Lower RBC

Higher CMC

Higher SII

Higher Hb

Lower CLC

Poor prognostic factors in UVA

(after treatment)

Higher PLR

Higher SII

Lower Hb

Higher NLR

Higher SII

Lower Hb

Lower RBC

Higher CLC

Higher PLT

Poor prognostic factors in MVA

(after treatment)

Higher PLT

Higher SII

Lower RBC

Lower WBC

Higher PLR

Lower RBC

Higher CLC

Higher PLT

OS, overall survival, DFS, disease-free-survival; UVA, univariate analysis; MVA, multivariate analysis; Hb, hemoglobin level; RetHb, RBC, red blood cells; WBC, white blood cells count; CLC, circulating lymphocyte count; CNC, circulating neutrophil count; CMC, circulating monocyte count; PLT, platelet cells count.

NLR, neutrophil/lymphocyte ratio; LMR, lymphocyte/monocyte ratio; PLR, platelet/lymphocyte ratio; SII, systemic immune inflammation index.

Table S9. Overall survival (OS) depending on post-tretament parameters in HPV-/HPV+ patients: univariate and multivariate analysis.

Variable

OS HPV-

OS HPV+

Univariate analysis

Multivariate analysis

Univariate analysis

Multivariate analysis

HR
(95%CI)

p-value

HR
(95%CI)

p-value

HR
(95%CI)

p-value

HR
(95%CI)

p-value

Hb 1 [g/dl] 
>10.1 vs. <10.1

0.27
(0.08-0.93)

0.037

0.47
(0.06-3.77)

0.478

RetHb 1 [/mm3] 
>37.8 vs. <37.8

1.37
(0.50-3.77)

0.537

0.67
(0.14-3.17)

0.613

RBC 1 [/mm3]
 >3.75 vs. <3.75

0.53
(0.24-1.17)

0.118

0.32
(0.12-0.86)

0.024

0.72
(0.23-2.29)

0.580

Ret 1 [/mm3] 
>27.5 vs. <27.5

4.15
(0.56-30.94)

0.166

11.24
(0.99-127.94)

0.051

0.48
(0.14-1.65)

0.245

WBC 1 [/mm3] 
>3.74 vs. <3.74

0.83
 (0.36-1.93)

0.664

0.16
(0.03-0.93)

0.042

2.60
(0.70-9.64)

0.152

CLC 1 [/mm3] 
>0.83 vs. <0.83

0.90
(0.36-2.27)

0.823

8.70
(2.26-33.54)

0.002

11.37
(2.61-49.64)

0.001

CNC 1 [/mm3] 
>2.32 vs. <2.32

1.15
(0.43-3.06)

0.782

7.48
(1.00-56.06)

0.050

1.51
(0.40-5.61)

0.541

CMC 1 [/mm3] 
>0.42 vs. <0.42

1.70
(0.68-4.27)

0.260

0.94
(0.30-2.90)

0.908

PLT 1 [/mm3] 
>313 vs. <313

2.06
(0.89-4.77)

0.092

3.37
(1.18-9.62)

0.023

2.05
(0.26-16.14)

0.495

NLR 1 
>6.45 vs. <6.45

1.15
(0.48-2.79)

0.754

0.43
(0.12-1.60)

0.209

MLR 1 
>0.63 vs. <0.63

0.85
(0.38-1.91)

0.697

0.35
(0.12-1.03)

0.057

0.36
(0.09-1.39)

0.138

PLR 1 
>404.17 vs. <404.17

2.55
(1.13-5.77)

0.024

0.87
(0.28-2.72)

0.815

SII 1
 >2763 vs. <2763

3.54
(1.28-9.80)

0.015

4.69
(1.23-17.97)

0.024

2.12
(0.46-9.81)

0.338

4.17
(0.76-22.88)

0.100

0, before treatment, Hb, hemoglobin level; RetHb, ret-hemoglobin level; RBC, red blood cells; Ret, reticulocyte count; WBC, white blood cells count; CLC, circulating lymphocyte count; CNC, circulating neutrophil count; CMC, circulating monocyte count; PLT, platelet cells count.

NLR, neutrophil/lymphocyte ratio; LMR, lymphocyte/monocyte ratio; PLR, platelet/lymphocyte ratio; SII, systemic immune inflammation index.

Table S10.  Disease-free survival (DFS) depending on post-treatent parameters in HPV-/HPV+ patients: univariate and multivariate analysis.

Variable

DFS HPV-

DFS HPV+

Univariate analysis

Multivariate analysis

Univariate analysis

Multivariate analysis

HR
(95%CI)

p-value

HR
(95%CI)

p-value

HR
(95%CI)

p-value

HR
(95%CI)

p-value

Hb 1 [g/dl] 
>10.8 vs. <10.8

0.37
(0.15-0.89)

0.027

2.35
(0.29-19.11)

0.424

RetHb 1 [/mm3] 
>38.0 vs. <38.0

2.28
(0.76-6.82)

0.142

0.72
(0.09-5.98)

0.761

RBC 1 [/mm3] 
>3.75 vs. <3.75

0.42
(0.19-0.93)

0.033

0.26
(0.11-0.66)

0.004

1.56
(0.31-7.71)

0.589

Ret 1 [/mm3] 
>27.5 vs. <27.5

4.03
(0.54-30.11)

0.174

0.39
(0.09-1.76)

0.221

WBC 1 [/mm3] 
>3.21 vs. <3.21

0.57
(0.23-1.44)

0.233

17780630

0.994

CLC 1 [/mm3] 
>0.37 vs. <0.37

0.56
(0.22-1.42)

0.226

4.03
(0.50-32.73)

0.193

CNC 1 [/mm3] 
>2.69 vs. <2.69

0.98
(0.40-2.35)

0.956

1.66
(0.40-6.95)

0.488

CMC 1 [/mm3] 
>0.47 vs. <0.47

0.50
(0.22-1.11)

0.087

1.31
(0.33-5.22)

0.706

PLT 1 [/mm3] 
>261 vs. <261

1.07
(0.48-2.39)

0.873

4.26
(1.06-17.07)

0.041

7.97
(1.55-41.00)

0.013

NLR 1 
>13.24 vs. <13.24

3.16
(1.18-8.50)

0.022

1.17
(0.14-9.52)

0.882

MLR 1 
>0.63 vs. <0.63

0.66
(0.29-1.51)

0.325

0.40
(0.15-1.03)

0.058

0.48
(0.10-2.38)

0.368

0.21
(0.04-1.31)

0.095

PLR 1 
>380 vs. <380

2.17
(0.95-4.99)

0.068

3.96
(1.46-10.77)

0.007

0.80
(0.20-3.18)

0.747

SII 1 
>2730 vs. <2730

3.25
(1.27-8.28)

0.014

1.33
(0.16-10.81)

0.790

0, before treatment, Hb, hemoglobin level; RetHb, ret-hemoglobin level; RBC, red blood cells; Ret, reticulocyte count; WBC, white blood cells count; CLC, circulating lymphocyte count; CNC, circulating neutrophil count; CMC, circulating monocyte count; PLT, platelet cells count.

NLR, neutrophil/lymphocyte ratio; LMR, lymphocyte/monocyte ratio; PLR, platelet/lymphocyte ratio; SII, systemic immune inflammation index.

  1. This study concludes that OS and DFS were significantly better in HPV+ patients compared to HPV- ones, there was no possible explanations in the discussion.

Answer: The explanation and comparison of these results has been added in the discussion as follows:

Generally, our study showed significantly better OS and DFS in HPV+ patients compared to HPV- ones (p = 0.0008 and p = 0.0009, respectively). Thus, HPV status was a very strong prognostic factor for OS and DFS. Survival of HPV+ patients was better despite of the higher regional disease advancement. These results are in full accordance in the literature data. Patients with HPV-related OPC have a better prognosis and longer survival compared to patients without HPV-related OPC with typical risk factors (smoking, alcohol abuse) [43,44]. A better prognosis is also observed in HPV+ patients with more advanced OPC with lymph nodes involvement [5,6]. Moreover, the HPV+ OPC is more responsive to radiotherapy (RT) and chemoradiotherapy (CRT) [4,5,45]. It allows for treatment de-escalation in HPV+ patients [45,46,47]. In our opinion, this strong impact of HPV status on survival was associated with the presence of different prognostic factors in HPV- and HPV patients. The better survival in HPV+ OPC patients is associated with a greater locoregional control, higher sensitivity to radiation or better radio-sensitization with the use of cisplatin [48]. The association between the superior survival of HPV+ OPC patients and the administered therapy is unclear. According to numerous authors, tumor HPV status is a strong and consistent determinant of better survival, regardless of treatment strategy (surgery, radiation therapy, concurrent CRT or induction chemotherapy plus concurrent CRT) with 5-year survival rates among HPV+ patients of approximately 75 to 80%, versus 45 to 50% among HPV- patients [48-52].

Round 2

Reviewer 1 Report

The authors have greatly improved the readability of this manuscript, but errors carried from previous version still can be found, including:

  1. p6 L9: ...G2 grading was reported in 34 (72.3%) and 21 (61.8%) patients in HPV- and HPV+ patients, respectively. G3 grading was more frequently noted in HPV+ patients compared to HPV- ones (14.9% vs.35.3%; ...)
  2. p6  L13: .... (71.2% vs. 47.1%, N2-3; p=0.003)
  3. p8 L2: (95%CI 0.99-5.47, p=0.053)
  4. The reviewer is still unable to validate all the numbers in Result Section 3.3. (p 8-9).  If the authors want to convince others by these data, please specify each sentence with their corresponding tables. Otherwise, Table 2 is much easier to understand than all the data described in p8-9.

Reviewer 2 Report

The authors have, or attempted to address all of my concerns. I am satisfied.

Reviewer 3 Report

I appreciate that the authors have addressed all the comments and gave responses very quickly. As I have mentioned before, the biggest concern is that the novelty of this study is low, as many studies have already demonstrated better overall survival of HPV-positive oropharyngeal patients. Second, the experiment's design is not rigorous, as seen from the result Table 1, the treatment strategy between HPV+ and HPV- patients are significantly different, so the treatment effect on the immune and hematological factors for OS and DFS is unclear, these questions need to be addressed.